# Recovery Analysis for Plug-and-Play Priors using the Restricted Eigenvalue Condition

**Jiaming Liu**
Washington University in St. Louis
jiaming.liu@wustl.edu

**M. Salman Asif**
University of California, Riverside
sasif@ece.ucr.edu

**Brendt Wohlberg**
Los Alamos National Laboratory
brendt@ieee.org

**Ulugbek S. Kamilov**
Washington University in St. Louis
kamilov@wustl.edu

## Abstract

The *plug-and-play priors (PnP)* and *regularization by denoising (RED)* methods have become widely used for solving inverse problems by leveraging pre-trained deep denoisers as image priors. While the empirical imaging performance and the theoretical convergence properties of these algorithms have been widely investigated, their recovery properties have not previously been theoretically analyzed. We address this gap by showing how to establish theoretical recovery guarantees for PnP/RED by assuming that the solution of these methods lies near the fixed-points of a deep neural network. We also present numerical results comparing the recovery performance of PnP/RED in compressive sensing against that of recent compressive sensing algorithms based on generative models. Our numerical results suggest that PnP with a pre-trained artifact removal network provides significantly better results compared to the existing state-of-the-art methods.

## 1 Introduction

Many imaging problems—such as denoising, inpainting, and super-resolution—can be formulated as an *inverse problem* involving the recovery of an image $x^* \in \mathbb{R}^n$ from noisy measurements

$$y = Ax^* + e \,, \tag{1}$$

where $A \in \mathbb{R}^{m \times n}$ is the measurement operator and $e \in \mathbb{R}^m$ is the noise. *Compressed sensing (CS)* [1, 2] is a related class of inverse problems that seek to recover a sparse vector $x^*$ from $m < n$ measurements. The sparse recovery is possible under certain assumptions on the measurement matrix, such as the *restricted isometry property (RIP)* [1] or the *restricted eigenvalue condition (REC)* [3, 4]. While traditional CS recovery relies on sparsity-promoting priors, recent work on *compressed sensing using generative models (CSGM)* [5] has broadened this perspective to priors specified through pre-trained generative models. CSGM has prompted a large amount of follow-up work on the design and theoretical analysis of algorithms that can leverage generative models as priors for image recovery [6–9].

*Plug-and-play priors (PnP)* [10, 11] and *regularization by denoising (RED)* [12] are two methods related to CSGM that can also leverage pre-trained deep models as priors for inverse problems. However, unlike CSGM, the regularization in PnP/RED is not based on restricting the solution to the range of a generative model, but rather on denoising the iterates with an existing *additive white Gaussian noise (AWGN)* removal method. The effectiveness of PnP/RED has been shown in a number of inverse problems [13–18], which has prompted researchers to investigate the theoretical properties and interpretations of PnP/RED algorithms [19–30].

35th Conference on Neural Information Processing Systems (NeurIPS 2021).

Despite the rich literature on both PnP/RED and CSGM, the conceptual relationship between these two classes of methods has never been formally investigated. In particular, while PnP/RED algorithms enjoy computational advantages over CSGM by not requiring nonconvex projections onto the range of a generative model, they lack theoretical recovery guarantees available for CSGM. In this paper, we address this gap by presenting the first recovery analysis of PnP/RED under the assumptions of CSGM. We show that if a measurement matrix satisfies a variant of REC from [5] over the range of a denoiser, then the distance of the PnP solutions to the true $\boldsymbol{x}^*$ can be explicitly characterized. We also present conditions under which the solutions of both PnP and RED coincide, providing sufficient conditions for the exact recovery of $\boldsymbol{x}^*$ using both methodologies. Our results highlight that the regularization in PnP/RED is achieved by giving preference to images near the *fixed points* of pre-trained deep neural networks. Besides new theory, this paper also presents numerical results directly comparing the recovery performance of PnP/RED against the recent algorithms in compressed sensing from random projections and subsampled Fourier measurements. These numerical results lead to new insights highlighting the excellent recovery performance of both PnP and RED, as well as the benefit of using priors specified as pre-trained *artifact removal (AR)* operators rather than AWGN denoisers.

All proofs and some technical details that have been omitted for space appear in the Supplement, which also provides more background and simulations. The code for our numerical evaluation is available at: `https://github.com/wustl-cig/pnp-recovery`.

## 2    Background

**Inverse problems.** A common approach to estimating $\boldsymbol{x}^*$ in (1) is to solve an optimization problem:

$$\min_{\boldsymbol{x} \in \mathbb{R}^n} g(\boldsymbol{x}) + h(\boldsymbol{x}) \quad \text{with} \quad g(\boldsymbol{x}) = \frac{1}{2}\|\boldsymbol{y} - \boldsymbol{A}\boldsymbol{x}\|_2^2 \;, \tag{2}$$

where $g$ is a data-fidelity term that quantifies consistency with the observed data $\boldsymbol{y}$ and $h$ is a regularizer that encodes prior knowledge on $\boldsymbol{x}$. For example, a widely-used regularizer in inverse problems is the nonsmooth *total variation (TV)* function $h(\boldsymbol{x}) = \tau\|\boldsymbol{D}\boldsymbol{x}\|_1$, where $\boldsymbol{D}$ is the gradient operator and $\tau > 0$ is the regularization parameter [31–33].

**Compressed sensing using generative models.** Generative priors have recently become popular for solving inverse problems [5], which typically require solving the optimization problem:

$$\min_{\boldsymbol{z} \in \mathbb{R}^k} \frac{1}{2}\|\boldsymbol{y} - \boldsymbol{A}\mathsf{W}(\boldsymbol{z})\|_2^2 \;, \tag{3}$$

where $\mathsf{W} : \mathbb{R}^k \to \mathsf{Im}(\mathsf{W}) \subseteq \mathbb{R}^n$ is a pre-trained generative model, such as StyleGAN-2 [34, 35]. The set $\mathsf{Im}(\mathsf{W})$ is the image set (or the range set) of the generator $\mathsf{W}$. In the past few years, several algorithms have been proposed for solving this optimization problem [6–9], including the recent algorithms *PULSE* [36] and *intermediate layer optimization (ILO)* [37] that can recover highly-realistic images. The recovery analysis of CSGM was performed under the assumption that $\boldsymbol{A}$ satisfies the *set-restricted eigenvalue condition (S-REC)* [5] over the range of the generative model:

$$\|\boldsymbol{A}\boldsymbol{x} - \boldsymbol{A}\boldsymbol{z}\|_2^2 \geq \mu\|\boldsymbol{x} - \boldsymbol{z}\|_2^2 - \eta \quad \forall \boldsymbol{x}, \boldsymbol{z} \in \mathsf{Im}(\mathsf{W}) \;, \tag{4}$$

where $\mu > 0$ and $\eta \geq 0$. S-REC implies that the pairwise distances between vectors in the range of the generative model must be well preserved in the measurement space. It thus broadens the traditional notions of REC and the *restricted isometry property (RIP)* in CS beyond sparse vectors [38].

**PnP and RED.** PnP [10, 11] refers to a family of iterative algorithms that are based on replacing the proximal operator $\mathsf{prox}_{\gamma h}$ of the regularizer $h$ within a proximal algorithm [39] by a more general denoiser $\mathsf{D} : \mathbb{R}^n \to \mathsf{Im}(\mathsf{D}) \subseteq \mathbb{R}^n$, such as BM3D [40] or DnCNN [41]. For example, the widely used *proximal gradient method (PGM)* [42–45] can be implemented as a PnP algorithm as [46]

$$\boldsymbol{x}^k = \mathsf{T}(\boldsymbol{x}^{k-1}) \quad \text{with} \quad \mathsf{T} := \mathsf{D}(\mathsf{I} - \gamma\nabla g) \;, \tag{5}$$

where $g$ is the data-fidelity term in (2), $\mathsf{I}$ denotes the identity mapping, and $\gamma > 0$ is the step size. Remarkably, this heuristic of using denoisers not associated with any $h$ within a proximal algorithm exhibited great empirical success [13–18] and spurred a great deal of theoretical work on PnP [19–28]. In particular, it has been recently shown in [23] that, when the residual of $\mathsf{D}$ is Lipschitz continuous, PnP-PGM converges to a point in the fixed-point set of the operator $\mathsf{T}$ that we denote $\mathsf{Fix}(\mathsf{T})$.

RED [12] is a related method, inspired by PnP, for integrating denoisers as priors for inverse problems. For example, the *steepest descent* variant of RED (SD-RED) [12] can be summarized as

$$\boldsymbol{x}^k = \boldsymbol{x}^{k-1} - \gamma\mathsf{G}(\boldsymbol{x}^{k-1}) \quad \text{with} \quad \mathsf{G} := \nabla g + \tau(\mathsf{I} - \mathsf{D}) , \tag{6}$$

where $\gamma > 0$ is the step size and $\tau > 0$ is the regularization parameter. For a locally homogeneous D that has a strongly passive and symmetric Jacobian, the solution of RED solves (2) with $h(\boldsymbol{x}) = (\tau/2)\boldsymbol{x}^{\mathsf{T}}(\boldsymbol{x} - \mathsf{D}(\boldsymbol{x}))$ [12, 22]. Subsequent work has resulted in a number of extensions of RED [29, 30, 47–49]. For example, it has been shown in [29] that, when D is a nonexpansive operator, SD-RED converges to a point in the zero set of operator G that we denote as $\mathsf{Zer}(\mathsf{G})$.

**Other related work.** While not directly related to our main theoretical contributions, it is worth briefly mentioning other important related families of algorithms that also use deep neural nets for regularizing ill-posed imaging inverse problems (see recent reviews of the area [50–53]). This work is most related to methods that rely on pre-trained priors that are integrated within iterative algorithms, such as a class of algorithms in compressive sensing known as *approximate message passing (AMP)* [54–57]. Another related family of algorithms are those based on the idea of *deep unrolling* (for an overview see Section IV-A in [53]). Inspired by LISTA [58], the unrolling algorithms interpret iterations of a regularized inversion as layers of a CNN and train it end-to-end in a supervised fashion [59–64]. Deep image prior [65] and deep decoder [66] also use neural networks as prior for images; instead of using a pre-trained generative network, they learn the parameters of the network while solving the inverse problem using the available measurements.

## 3 Recovery Analysis for PnP and RED

We present two sets of theoretical results for PnP-PGM (5) using the measurement model (1) and the least-squares data-fidelity term (2). We first establish recovery bounds for PnP under a set of sufficient conditions, and then address the relationship between the solutions of PnP and RED. The proofs of all the theorems will be provided in the Supplement. We start by discussing two assumptions that serve as sufficient conditions for our analysis of PnP.

**Assumption 1.** *The residual* $\mathsf{R} := \mathsf{I} - \mathsf{D}$ *of the operator* D *is bounded by* $\delta$ *and Lipschitz continuous with constant* $\alpha > 0$*, which can be written as*

$$\|\mathsf{R}(\boldsymbol{x})\|_2 \le \delta \quad \text{and} \quad \|\mathsf{R}(\boldsymbol{x}) - \mathsf{R}(\boldsymbol{z})\|_2 \le \alpha\|\boldsymbol{x} - \boldsymbol{z}\|_2, \quad \forall \boldsymbol{x}, \boldsymbol{z} \in \mathbb{R}^n . \tag{7}$$

The rationale for stating Assumption 1 in terms of the residual R is based on our interest in *residual* deep neural nets that take a noisy or an artifact-corrupted image at the input and produce the corresponding noise or artifacts at the output. The success of residual learning in the context of image restoration is well known [41]. Prior work has also shown that Lipschitz constrained residual networks yield excellent performance without sacrificing stable convergence [23, 29].

Related assumptions have been used in earlier convergence results for PnP [19, 23]. For example, one of the most-widely known PnP convergence results relies on the boundedness of D [19]. The Lipschitz continuity of the residual R has been used in the recent analysis of several PnP algorithms in [23]. Both of these assumptions are relatively easy to implement for deep priors. For example, the boundedness of R can be enforced by simply bounding each output pixel to be within $[0, \nu]$ for images in $[0, \nu]^n \subset \mathbb{R}^n$ for some $\nu > 0$. The $\alpha$-Lipschitz continuity of R can be enforced by using any of the recent techniques for training Lipschitz constrained deep neural nets [23, 67–69]. Fig. 1 presents an empirical evaluation of the Lipschitz continuity of R used in our simulations.

**Assumption 2.** *The measurement operator* $\boldsymbol{A} \in \mathbb{R}^{m \times n}$ *satisfies the set-restricted eigenvalue condition (S-REC) over* $\mathsf{Im}(\mathsf{D}) \subseteq \mathbb{R}^n$ *with* $\mu > 0$*, which can be written as*

$$\|\boldsymbol{A}\boldsymbol{x} - \boldsymbol{A}\boldsymbol{z}\|_2^2 \ge \mu\|\boldsymbol{x} - \boldsymbol{z}\|_2^2, \quad \forall \boldsymbol{x}, \boldsymbol{z} \in \mathsf{Im}(\mathsf{D}) . \tag{8}$$

S-REC in Assumption 2 was adopted from the corresponding assumption for CSGM stated in (4), which establishes a natural conceptual link between those two classes of methods. The main limitation of Assumption 2, which is also present in the traditional RIP/REC assumptions for compressive sensing, lies in the difficulty of verifying it for a given measurement operator $\boldsymbol{A}$. There has been significant activity in investigating the validity of related conditions for randomized matrices for different classes of signals [4, 70–72], including for those synthesized by generative models [5, 6, 9, 37].

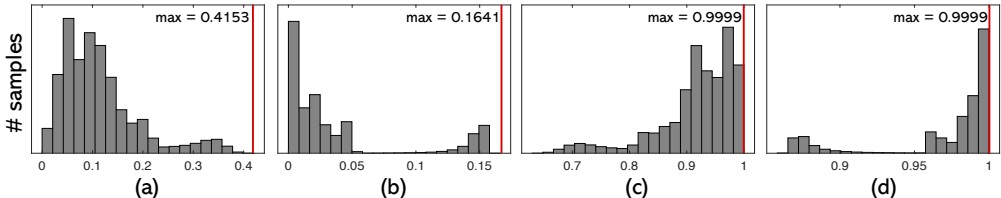

Figure 1: *Empirical evaluation of the Lipschitz continuity of R and D used in our simulations and stated in Assumptions 1 and 3. As described in the main text, we trained two types of Lipschitz constrained networks, where the first simply denoises AWGN and the second removes artifacts specific to the PnP iterations. (a) and (b) show the histograms of $\|R(x) - R(z)\|_2/\|x - z\|_2$ for the denoiser and the artifact-removal operator, respectively. (c) and (d) show the histograms of $\|D(x) - D(z)\|_2/\|x - z\|_2$ for the same two operators. Note the empirical nonexpansiveness of D despite the fact that Lipschitz continuity was only imposed on the residual R during training.*

Despite this limitation, Assumption 2 is still conceptually useful as it allows us to relax the strong convexity assumption used in the convergence analysis in [23] by stating that it is sufficient for the strong convexity to hold *only* over the image set $\mathsf{Im}(D)$ rather than over the whole $\mathbb{R}^n$. This suggests a new research direction for PnP on designing deep priors with range spaces restricted to satisfy S-REC for some $A$. In the Supplement, we present an empirical evaluation of $\mu$ for the measurement operators used in our experiments by sampling from $\mathsf{Im}(D)$.

Consider the set $\mathsf{Fix}(D) \coloneqq \{x \in \mathbb{R}^n : x = D(x)\}$ of the *fixed points* of D. Note that $\mathsf{Fix}(D)$ is equivalent to the set $\mathsf{Zer}(R) \coloneqq \{x \in \mathbb{R}^n : R(x) = \mathbf{0}\}$ of the *zeros* of the residual $R = I - D$. Intuitively, $\mathsf{Zer}(R)$ consists of all images that produce no residuals, and therefore can be interpreted as the set of all *noise-free* images according to the network. Similarly, when R is trained to predict artifacts in an image, $\mathsf{Zer}(R)$ is the set of images that are *artifact-free* according to R. In the subsequent analysis, we use the notation $\mathsf{Zer}(R)$, but these results can be equivalently stated using $\mathsf{Fix}(D)$.

We first state the PnP recovery in the setting where there is no noise and $x^* \in \mathsf{Zer}(R)$.

**Theorem 1.** *Run PnP-PGM for $t \geq 1$ iterations under Assumptions 1-2 for the problem* (1) *with no noise and $x^* \in \mathsf{Zer}(R)$. Then, the sequence $x^t$ generated by PnP-PGM satisfies*

$$\|x^t - x^*\|_2 \leq c\|x^{t-1} - x^*\|_2 \leq c^t\|x^0 - x^*\|_2 , \tag{9}$$

*where $x^0 \in \mathsf{Im}(D)$ and $c \coloneqq (1 + \alpha) \max\{|1 - \gamma\mu|, |1 - \gamma\lambda|\}$ with $\lambda \coloneqq \lambda_{\max}(A^\mathsf{T}A)$.*

The proof of the theorem is available in the Supplement. Theorem 1 extends the theoretical analysis of PnP in [23] by showing convergence to the true solution $x^*$ of (1) instead of the fixed points $\mathsf{Fix}(T)$ of T in (5). The condition $x^0 \in \mathsf{Im}(D)$ can be easily enforced by simply passing any initial image through the operator D. One does not necessarily need $\alpha < 1$, for the convergence result in Theorem 1. As shown in [23], the coefficient $c$ in Theorem 1 is less than one if

$$\frac{1}{\mu(1 + 1/\alpha)} < \gamma < \frac{2}{\lambda} - \frac{1}{\lambda(1 + 1/\alpha)} , \tag{10}$$

which is possible if $\alpha < 2\mu/(\lambda - \mu)$. Since all PnP algorithms have the same fixed points [20, 24], our result implies that PnP can exactly recover the true solution $x^*$ to the inverse problem, which extends the existing theory in the literature that only shows convergence to $\mathsf{Fix}(T)$.

We now present a more general result that relaxes the assumptions in Theorem 1.

**Theorem 2.** *Run PnP-PGM for $t \geq 1$ iterations under Assumptions 1-2 for the problem* (1) *with $x^* \in \mathbb{R}^n$ and $e \in \mathbb{R}^m$. Then, the sequence $x^t$ generated by PnP-PGM satisfies*

$$\|x^t - x^*\|_2 \leq c\|x^{t-1} - x^*\|_2 + \varepsilon \leq c^t\|x^0 - x^*\|_2 + \frac{\varepsilon(1 - c^t)}{(1 - c)} , \tag{11}$$

*where $x^0 \in \mathsf{Im}(D)$ and*

$$\varepsilon \coloneqq (1 + c)\left[\left(1 + 2\sqrt{\lambda/\mu}\right)\|x^* - \mathsf{proj}_{\mathsf{Zer}(R)}(x^*)\|_2 + 2/\sqrt{\mu}\|e\|_2 + \delta(1 + 1/\alpha)\right] \tag{12}$$

*and $c \coloneqq (1 + \alpha) \max\{|1 - \gamma\mu|, |1 - \gamma\lambda|\}$ with $\lambda \coloneqq \lambda_{\max}(A^\mathsf{T}A)$.*

Theorem 2 extends Theorem 1 by allowing $\boldsymbol{x}^*$ to be outside of $\mathsf{Zer}(\mathsf{R})$ and extends the analysis in [5] by considering operators $\mathsf{D}$ that do not necessarily project onto the range of a generative model. In the error bound $\varepsilon$, the first two terms are the distance of $\boldsymbol{x}^*$ to $\mathsf{Zer}(\mathsf{R})$ and the magnitude of the error $\boldsymbol{e}$, and have direct analogs in standard compressed sensing. The third term is the consequence of the possibility for the solution of PnP not being in the zero-set of R and one can show that when $\mathsf{Zer}(\mathsf{R}) \cap \mathsf{Zer}(\nabla g) \neq \varnothing$, then the third term disappears. As reported in the Supplement, we empirically verified that the distance of the PnP solution to $\mathsf{Zer}(\mathsf{R})$ is small for both the denoiser and the artifact-removal operators used in our experiments.

Our final result explicitly relates the solutions of PnP and RED. In order to obtain the result, we need an additional assumption that the denoiser $\mathsf{D} = \mathsf{I} - \mathsf{R}$ is nonexpansive.

**Assumption 3.** *The denoiser* $\mathsf{D}$ *is nonexpansive*

$$\|\mathsf{D}(\boldsymbol{x}) - \mathsf{D}(\boldsymbol{z})\|_2 \leq \|\boldsymbol{x} - \boldsymbol{z}\|_2, \quad \forall \boldsymbol{x}, \boldsymbol{z} \in \mathbb{R}^n .$$

This is related but different from Assumption 1 that assumes the residual R is $\alpha$-Lipschitz continuous.

The convergence of SD-RED in (6) to $\mathsf{Zer}(\mathsf{G})$ can be established for a nonexpansive operator $\mathsf{D}$ [29]. In principle, the nonexpansiveness of $\mathsf{D}$ can be enforced during the training of the prior in the same manner as that of the more general Lipschitz continuity. However, the prior in our numerical evaluations is trained to have a contractive residual R without any explicit constraints on D. As a reminder, the nonexpansiveness of R is only a necessary (but not sufficient) condition for the nonexpansiveness of D [73]. Despite this fact, our empirical evaluation of the Lipschitz constant of D in Fig. 1 indicates that D used in our experiments is nonexpansive.

**Theorem 3.** *Suppose that Assumptions 1-3 are satisfied and that* $\mathsf{Zer}(\nabla g) \cap \mathsf{Zer}(\mathsf{R}) \neq \varnothing$, *then PnP and RED have the same set of solutions:* $\mathsf{Fix}(\mathsf{T}) = \mathsf{Zer}(\mathsf{G})$.

As a reminder, the solutions of PnP correspond to the fixed-points of the operator T defined in (5), while those of RED to the zeroes of the operator G defined in (6). The assumption that $\mathsf{Zer}(\nabla g) \cap \mathsf{Zer}(\mathsf{R}) \neq \varnothing$ implies that there exist vectors that are noise/artifact free according to R and consistent with the measurements $\boldsymbol{y}$. While this assumption is not universally applicable to all the inverse problems and priors, it still provides a highly-intuitive sufficient condition for the PnP/RED equivalence. Although the relationship between PnP and RED has been explored in the prior work [22, 30], to the best of our knowledge, Theorem 3 is the first to prove explicit equivalence. If one additionally considers PnP-PGM with a step size that satisfies the condition in (10), then T is a contraction over $\mathsf{Im}(\mathsf{D})$, which implies that PnP-PGM converges linearly to its unique fixed point in $\mathsf{Im}(\mathsf{D})$. The direct corollary of our analysis is that, in the noiseless scenario $\boldsymbol{y} = \boldsymbol{A}\boldsymbol{x}^*$ with $\boldsymbol{x}^* \in \mathsf{Zer}(\mathsf{R})$, the image $\boldsymbol{x}^*$ is the unique fixed point of both PnP and RED over $\mathsf{Im}(\mathsf{D})$.

It is worth mentioning that several of our assumptions have been stated in a way that simplifies mathematical exposition, but can alternatively be presented in a significantly weaker form. For example, the boundedness assumption in (1)—which is used only in the proof of Theorem 2—does not have to hold everywhere, but only at the fixed points of PnP. Indeed, as can be seen in eq. (6) of the Supplement, the constant $\delta$ is only used to bound the norm of the residual at the fixed-point of PnP-PGM. Similarly, we do not need the nonexpansiveness of D in Assumption 3 to be true everywhere, but only at the fixed points of RED-SD (see page 4 of the Supplement).

In summary, our theoretical analysis reveals that the fixed-point convergence of PnP/RED algorithms can be strengthened to provide recovery guarantees when S-REC from CSGM is satisfied. Since PnP/RED algorithms do not require nonconvex projections onto the range of a generative model, they enjoy computational benefits over methods that use generative models as priors. However, the literature on generative models is rich with theoretical bounds and recovery guarantees compared to that of PnP/RED. We believe that our work suggests an exciting new direction of research for PnP/RED by showing that a similar analysis can be carried out for PnP/RED.

## 4 Numerical Evaluation

Before presenting our numerical results, it is important to note that PnP and RED are well-known methods and it is *not* our aim to claim any algorithmic novelty with respect to them. However, comparing PnP/RED to state-of-the-art *compressed sensing (CS)* algorithms is of interest in the

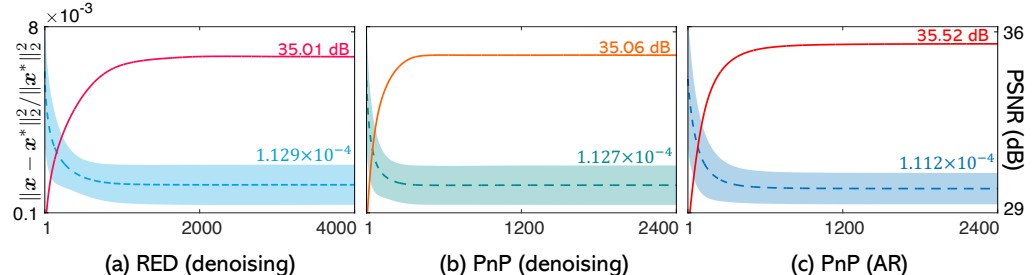

(a) RED (denoising)   (b) PnP (denoising)   (c) PnP (AR)

Figure 2: *Empirical evaluation of the convergence of PnP/RED to the true solution $x^*$ using a denoiser and an artifact removal (AR) operator. Average normalized distance and PSNR relative to the true solution $x^*$ are plotted with the shaded areas representing the range of values attained over all test images. Note the similar recovery performance of PnP and RED, as well as the improvement in performance due to a prior trained to remove artifacts specific to PnP iterations (rather than an AWGN denoiser).*

Table 1: Numerical evaluation of the CS recovery in terms of PSNR (dB) on BSD68 and Set11.

| CS Ratio / Method | BSD68 | | | | Set11 | | | |
|---|---|---|---|---|---|---|---|---|
| | 10% | 30% | 40% | 50% | 10% | 30% | 40% | 50% |
| TV | 24.56 | 28.61 | 30.27 | 31.98 | 24.47 | 30.21 | 32.29 | 34.27 |
| SDA [76] | 23.12 | 26.38 | 27.41 | 28.35 | 22.65 | 26.63 | 27.79 | 28.95 |
| ReconNet [77] | 24.15 | 27.53 | 29.08 | 29.86 | 24.28 | 28.74 | 30.58 | 31.50 |
| ISTA-Net [62] | 25.02 | 29.93 | 31.85 | 33.61 | 25.80 | 32.91 | 35.36 | 37.43 |
| ISTA-Net$^+$ [62] | 25.33 | 30.34 | 32.21 | 34.01 | 26.64 | 33.82 | 36.06 | 38.07 |
| RED (denoising) | 24.97 | 30.20 | 32.25 | 34.39 | 27.70 | 35.01 | 37.28 | 39.26 |
| PnP (denoising) | 25.06 | 30.31 | 32.29 | 34.35 | 27.76 | 35.06 | 37.30 | 39.21 |
| PnP (AR) | **26.46** | **31.33** | **33.18** | **34.92** | **28.98** | **35.53** | **37.34** | **39.29** |

context of our theory. Our goal in this section is thus to both (a) empirically evaluate the recovery performance of PnP/RED and (b) compare their performances relative to widely-used CS algorithms.

We consider two scenarios: *(a) CS using random projections* and *(b) CS for magnetic resonance imaging (CS-MRI)*. In order to gain a deeper insights into performance under subsampling, we use an idealized noise-free setting; however, we expect similar relative performances under noise. For each scenario, we include comparisons with several well-established methods based on deep learning.

We consider two priors for PnP/RED: (i) an AWGN denoiser and (ii) an *artifact-removal (AR)* operator trained to remove artifacts specific to the PnP iterations. We implement both priors[1] using the DnCNN architecture [41], with its batch normalization layers removed for controlling the Lipschitz constant of the network via spectral normalization [68]. We train the denoiser as a nonexpansive residual network R that predicts the noise residual from a noisy input image. Thus, R satisfies the necessary condition for the nonexpansiveness of D. Similar to [74], we train the AR prior by including it into a *deep unfolding* architecture that performs PnP iterations. When equipped with spectral normalization [68], the residual R of the AR operator still satisfies Lipschitz continuity assumptions and achieves superior performance compared to the denoiser (as corroborated by our results). Our implementation also relies on the scaling strategy from [75] for controlling the influence of D relative to $g$. The reconstruction quality is quantified using the peak signal-to-noise ratio (PSNR) in dB.

## 4.1 Reconstruction of Natural Images from Random Projections

We adopt a simulation setup widely-used in the CS literature, in which non-overlapping $33 \times 33$ patches of an image are measured using the same $m \times n$ random Gaussian matrix $A$, whose rows have been orthogonalized [62, 77]. The patches are vectorized to $n = 1089$-length vectors $x^*$. The training data for the denoiser is generated by adding AWGN to the images from the BSD500 [78] and

---

[1]For additional details and code see the Supplement and the GitHub repository.

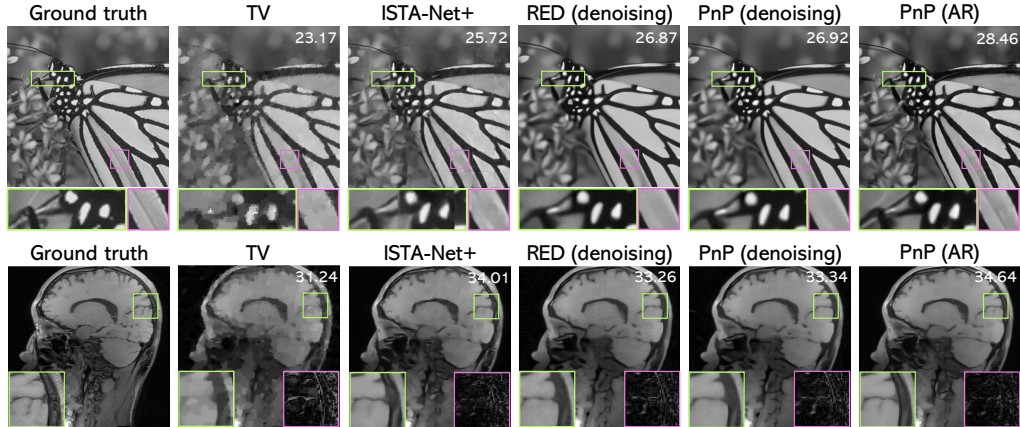

Figure 3: *Visual evaluation of various compressive sensing algorithms at* $10\%$ *sampling on two imaging problems: (top) reconstruction of* Butterfly *from Set11; (bottom) reconstruction of a brain MR image from its radial Fourier measurements. The pink box in the bottom image provides the error residual that was amplified by* $10\times$ *for better visualization. Note the similar performance of PnP and RED, as well as the competitiveness of both relative to other methods. Additionally, note the improvement due to the usage of an AR prior instead of an AWGN denoiser within PnP.*

Table 2: Average PSNR values for various CS-MRI methods on test images from [62].

| CS Ratio
Method | 10% | 20% | 30% | 40% | 50% |
|---|---|---|---|---|---|
| **TV** | 31.36 | 35.62 | 38.41 | 40.43 | 42.20 |
| **ADMM-Net [61]** | 34.19 | 37.17 | 39.84 | 41.56 | 43.00 |
| **ISTA-Net$^{+}$ [62]** | 34.65 | 38.70 | 40.97 | 42.65 | 44.12 |
| **RED (denoising)** | 34.37 | 38.63 | 40.94 | 42.62 | 44.21 |
| **PnP (denoising)** | 34.56 | 38.74 | 41.06 | 42.73 | 44.24 |
| **PnP (AR)** | **35.21** | **39.05** | **41.28** | **42.96** | **44.47** |

DIV2K datasets [79]. We pre-train several deep models as denoisers for $\sigma \in [1, 15]$, using $\sigma$ intervals of 0.5, and use the denoiser achieving the best PSNR value in each experiment. We use the same set of 91 images as in [77] to train the AR operators that are implemented on individual image patches at a time for the CS ratios $(m/n)$ of $\{10\%, 30\%, 40\%, 50\%\}$. In order to overcome the block-artifacts in the recovered images, we implement PnP and RED regularizers over the entire image while still using the per-patch measurement model for $\nabla g$.

Our first numerical study in Fig. 1 evaluates the Lipschitz continuity of our pre-trained denoisers and the AR operators by following the procedure in [23]. We use the residual R and its corresponding operator $D = I - R$ and plot the histograms of $\alpha_1 = \|R(\boldsymbol{x}) - R(\boldsymbol{z})\|_2 / \|\boldsymbol{x} - \boldsymbol{z}\|_2$ and $\alpha_2 = \|D(\boldsymbol{x}) - D(\boldsymbol{z})\|_2 / \|\boldsymbol{x} - \boldsymbol{z}\|_2$ over 1160 AWGN corrupted image pairs extracted from BSD68. The maximum value of each histogram is indicated by a vertical bar, providing an empirical bound on the Lipschitz constants. Fig. 1 confirms empirically that both R and D are contractive operators.

Theorem 2 establishes that the sequence of iterates $\boldsymbol{x}^t$ generated by PnP-PGM converges to the true solutions $\boldsymbol{x}^*$ up to an error term. Fig. 2 illustrates the convergence behavior of PnP/RED in terms of $\|\boldsymbol{x}^t - \boldsymbol{x}^*\|_2^2 / \|\boldsymbol{x}^*\|_2^2$ and peak signal-to-noise ratio (PSNR) for CS with subsampling ratio of $30\%$ on Set11. The shaded areas represent the range of values attained across all test images. The results in Fig. 2 are consistent with our general observation that the PnP/RED algorithms converge in all our experiments for both types of priors and achieve excellent recovery performance.

We also report the average PSNR values obtained by five baseline CS algorithms, namely TV [33], SDA [76], ReconNet [77], ISTA-Net [62] and ISTA-Net+ [62]. TV is an iterative methods that does not require training, while the other four are all deep learning-based methods that have publicly available implementations. The numerical results on Set11 and BSD68 with respect to four measure-

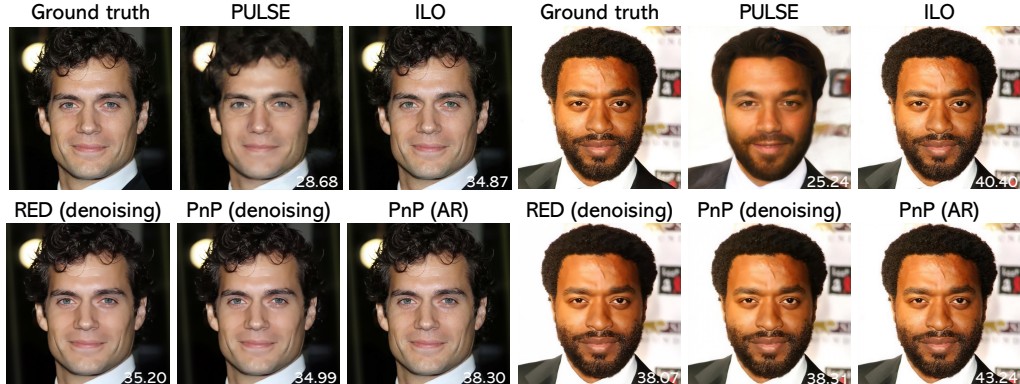

Figure 4: *Visual evaluation of PnP/RED and two methods using generative models as priors on the CelebA HQ [82] dataset at 10% CS sampling. Note the visual and quantitative similarity of PnP and RED when both are using AWGN denoisers. PnP using an artifact-removal (AR) prior visually matches the performance of ILO based on StyleGAN2, which highlights the benefit of using pre-trained AR operators within PnP. Best viewed by zooming in the display.*

Table 3: Average PSNR (dB) values for several algorithms on test images from CelebA HQ.

| CS Ratio
Method | 10% | 20% | 30% | 40% | 50% |
|---|---|---|---|---|---|
| **TV** | 32.13 | 35.24 | 37.41 | 39.35 | 41.29 |
| **PULSE [36]** | 27.45 | 29.98 | 33.06 | 34.25 | 34.77 |
| **ILO [37]** | 36.15 | 40.98 | 43.46 | 47.89 | 48.21 |
| **RED (denoising)** | 35.46 | 41.59 | 45.65 | 48.13 | 52.17 |
| **PnP (denoising)** | 35.61 | 41.51 | 45.71 | 48.05 | 52.24 |
| **PnP (AR)** | **39.19** | **44.20** | **48.66** | **51.32** | **53.89** |

ment rates are summarized in Table 1. We observe that the performances of PnP and RED are nearly identical to one another. The result also highlights that PnP using the AR prior provides the best performance[2] compared to all the other methods, outperforming PnP using the AWGN denoiser by at least 0.57 dB on BSD68. Fig. 3 (top) shows visual examples for an image from Set11. Note that both PnP and RED yield similar visual recovery performance. The enlarged regions in the image suggest that PnP (AR) better recovers the fine details and sharper edges compared to other methods.

## 4.2 Image reconstruction in Compressed Sensing MRI

MRI is a widely-used medical imaging technology that has known limitations due to the low speed of data acquisition. CS-MRI [80, 81] seeks to recover an image $x^*$ from its sparsely-sampled Fourier measurements. We simulate a single-coil CS-MRI using radial Fourier sampling. The measurement operator $A$ is thus $A = PF$, where $P$ is the diagonal sampling matrix and $F$ is the Fourier transform.

The priors for PnP/RED were trained using the brain dataset from [62], where the test set contains 50 slices of $256 \times 256$ images (i.e., $n = 65536$). We train seven variants of DnCNN, each using a separate noise level from $\sigma \in \{1, 1.5, 2, 2.5, 3, 4, 5\}$. Similarly, we separately train the AR operators for different CS ratios $(m/n)$, initializing the weights of the models from the pre-trained denoiser with $\sigma = 2$. For these sets of experiments, we also equipped PnP/RED with *Nesterov acceleration* [83] for faster convergence. We compare PnP/RED against publicly available implementations of several well-known methods, including TV [33], ADMM-Net [61], and ISTA-Net[+] [62]. The last two are deep unrolling methods that train both image transforms and shrinkage functions within the algorithm.

Table 2 reports the results for five CS ratios. The visual comparison can be found in Fig. 3 (bottom). It can been seen that PnP/RED with an AWGN denoiser matches the performance of ISTA-Net[+]

---

[2]We did not use RED with the AR prior in our experiments since it is expected to closely match PnP.

and outperforms ADMM-Net at higher sampling ratios, while PnP with an AR prior improves over PnP/RED with an AWGN denoiser [84]. Note also the similarity of PnP and RED performances.

### 4.3 Comparison with generative models on human faces

We numerically evaluated the recovery performance of PnP/RED in CS against two recent algorithms using generative models: PULSE [36] and ILO [37]. Similar to the measurement matrix used for grayscale images, we use orthogonalized random Gaussian matrices for sampling image blocks of size $33 \times 33 \times 3$. The test images correspond to 15 images randomly selected from CelebA HQ [82] dataset, each of size $1024 \times 1024$ pixels. We use the DIV2K [79] and 200 high quality face images from FFHQ dataset [34] to train the PnP/RED denoisers for color image denoising at six noise levels corresponding to $\sigma \in \{1, 2, 3, 4, 7, 10\}$. We use the same training set to train the AR operators for CS ratios of $[10\%, 50\%]$, using the ratio intervals of $10\%$. Similar to CS-MRI, we equipped PnP/RED with Nesterov acceleration. The PSNR comparison between different methods is presented in Table 3. It can be seen that ILO outperforms PULSE in terms of PSNR, which is consistent with the results in [37]. Note also how PnP/RED match or sometimes quantitively outperform ILO at high CS ratios, with PnP (AR) leading to significantly better results compared to PnP (denoising). Fig. 4 provides visual reconstruction examples. Note the ILO images are sharper compared to PnP/RED with denoisers because ILO uses a state-of-the-art generative model specifically trained on face images. However, PnP (AR) achieves better PSNR and a similar visual quality as ILO.

## 5 Conclusion and Future Work

The main goal of this work is to address the theoretical gap between two-widely used classes of methods for solving inverse problems, namely PnP/RED and CSGM. Motivated by the theoretical analysis of CSGM, we used S-REC to establish recovery guarantees for PnP/RED. Our theoretical results provide a new type of convergence for PnP-PGM that goes beyond a simple fixed-point convergence by showing convergence relative to the true solution. Additionally, we show the full equivalence of PnP and RED under some explicit conditions on the inverse problem. While the focus of this work is mainly theoretical, we presented several numerical evaluations that can provide additional insights into PnP/RED and their performance relative to standard methods used in compressed sensing. Empirically, we observed the similarity of PnP/RED in image reconstruction from subsampled random projections and Fourier transform. We also provided additional evidence on the suboptimality of AWGN denoisers compared to artifact-removal operators that take into account the actual artifacts within PnP iterates.

The work presented in this paper has a certain number of limitations and possible directions for improvement. The main limitation of our theoretical analysis, which is common to all compressive sensing research, is in the difficulty of theoretically verifying S-REC for a given measurement operator. One can also consider the Lipschitz assumptions on R/D as a limitation, since those can have a negative impact on the recovery. However, our results suggest that even with Lipschitz constrained priors, PnP/RED are competitive with widely-known CS algorithms. While PnP/RED can be implemented using non-Lipschitz-constrained priors, we expect that this will reduce their stability and ultimately hurt their recovery performances. A relatively minor limitation of our simulations is that they were performed without AWGN. One can easily re-run our code by including AWGN and we expect that the relative performances will be preserved for a reasonable amount of noise. We hope that this work will inspire further theoretical and algorithmic research on PnP/RED that will lead to extensions and improvements to our results.

## 6 Broader impact

This work is expected to impact the area of imaging inverse problems with potential applications to computational microscopy, computerized tomography, medical imaging, and image restoration. There is a growing need in imaging to deal with noisy and incomplete measurements by integrating multiple information sources, including physical information describing the imaging instrument and learned information characterizing the statistical distribution of the desired image. The ability to accurately solve inverse problems has the potential to enable new technological advances for imaging. These advances might lead to new imaging tools for diagnosing health conditions, understanding

biological processes, or inferring properties of complex materials. Traditionally, imaging relied on linear models and fixed transforms (filtered back projection, wavelet transform) that are relatively straightforward to understand. Learning based methods, including PnP and RED, have the potential to enable new technological capabilities; yet, they also come with a downside of being much more complex. Their usage might thus lead to unexpected outcomes and surprising results when used by non-experts. While we aim to use our method to enable positive contributions to humanity, one can also imagine nonethical usage of imaging technology. This work focuses on understanding theoretical properties of imaging algorithms using learned priors, but it might be adopted within broader data science, which might lead to broader impacts that we have not anticipated.

## Acknowledgments and Disclosure of Funding

Research presented in this article was supported by NSF awards CCF-1813910, CCF-2043134, and CCF-2046293 and by the Laboratory Directed Research and Development program of Los Alamos National Laboratory under project number 20200061DR.

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
