# Supplementary Material for "Recovery Analysis for Plug-and-Play Priors using the Restricted Eigenvalue Condition"

**Jiaming Liu**
Washington University in St. Louis
jiaming.liu@wustl.edu

**M. Salman Asif**
University of California, Riverside
sasif@ece.ucr.edu

**Brendt Wohlberg**
Los Alamos National Laboratory
brendt@ieee.org

**Ulugbek S. Kamilov**
Washington University in St. Louis
kamilov@wustl.edu

The mathematical analysis presented in this supplementary document builds on two distinct lines of work: (a) monotone operator theory [1, 2] and (b) compressive sensing using generative models (CSGM) [3]. In Section A, we build on past work to prove the convergence of PnP-PGM to the true solution of the inverse problem in the absence of noise. In Section B, we extend the result in Section A to $x^* \in \mathbb{R}^n$ and $e \in \mathbb{R}^m$ (i.e., when the signal can be arbitrary and measurements can have noise). In Section C, we show that PnP/RED can have the same set of solutions under some specific conditions. In Section D, we provide background material useful for our theoretical analysis. Finally, in Section E, we provide additional technical details on our implementations and simulations omitted from the main paper due to space.

The algorithmic details of PnP-PGM and SD-RED are summarized in Fig. 1. It is important to note that it is not our intent to claim any algorithmic novelty in PnP/RED, which are well-known methods. However, there is a strong interest in understanding the theoretical properties of PnP/RED in terms of both recovery and convergence. The main contribution of this work is the development of new theoretical insights into the recovery and convergence of PnP/RED. Finally, our code, including pre-trained denoisers and AR operators, is also included in the supplementary material.

We follow the same notation in the supplement as in the main manuscript. The measurement model corresponds to $y = Ax^* + e$, where $x^*$ is the true solution and $e$ is the noise. The function $g(x) = \frac{1}{2}\|y - Ax\|_2^2$ denotes the data-fidelity term. The operator D denotes the PnP/RED prior, which is implemented via its residual $\mathsf{R} := \mathsf{I} - \mathsf{D}$. The operator $\mathsf{T} := \mathsf{D}(\mathsf{I} - \gamma\nabla g)$ denotes the PnP update and $\mathsf{G} := \nabla g + \tau\mathsf{R}$ denotes the term used to compute RED updates.

## A Proof of Theorem 1

In this section, we prove the first of the main theoretical result in this work, namely the convergence of PnP-PGM to the true solution of the problem $y = Ax^*$ when $x^* \in \mathsf{Zer}(\mathsf{R})$. Our analysis extends the existing convergence analysis of PnP-PGM from [4], which proved a linear convergence of the algorithm to $\mathsf{Fix}(\mathsf{T})$. Here we extend [4] by using the fact that $x^* \in \mathsf{Zer}(\mathsf{R}) \cap \mathsf{Zer}(\nabla g)$ and relaxing the assumption of strong convexity in [4] to S-REC over $\mathsf{Im}(\mathsf{D})$.

**Theorem 1.** *Run PnP-PGM for $t \geq 1$ iterations under Assumptions 1-2 for the problem (1) of the main paper with no noise and $x^* \in \mathsf{Zer}(\mathsf{R})$. Then, the sequence $x^t$ generated by PnP-PGM satisfies*

$$\|x^t - x^*\|_2 \leq c\|x^{t-1} - x^*\|_2 \leq c^t\|x^0 - x^*\|_2 \,, \tag{1}$$

*where $x^0 \in \mathsf{Im}(\mathsf{D})$ and $c := (1 + \alpha)\max\{|1 - \gamma\mu|, |1 - \gamma\lambda|\}$ with $\lambda := \lambda_{\max}(A^\mathsf{T}A)$.*

35th Conference on Neural Information Processing Systems (NeurIPS 2021).

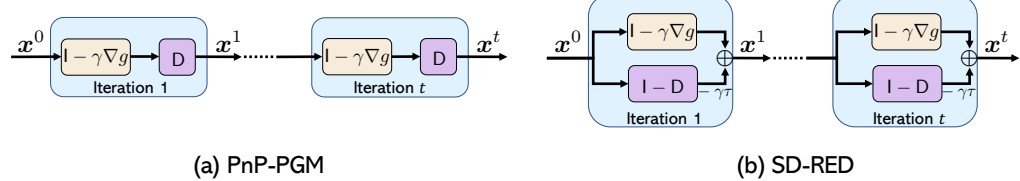

(a) PnP-PGM                      (b) SD-RED

Figure 1: *Algorithmic details of two optimization methods used in this work: (a) PnP-PGM and (b) SD-RED. Both algorithms are initialized with $\boldsymbol{x}^0$ and perform $t \geq 1$ iterations.*

Suppose all the assumptions for Theorem 1 are true and the step size $\gamma > 0$ is selected in a way that satisfies eq. (10) in the main paper. First note that we have assumed that $\boldsymbol{x}^0 \in \mathsf{Im}(\mathsf{D})$ and we have

$$\boldsymbol{x}^t = \mathsf{T}(\boldsymbol{x}^{t-1}) = \mathsf{D}(\boldsymbol{x}^{t-1} - \gamma \nabla g(\boldsymbol{x}^{t-1})) \in \mathsf{Im}(\mathsf{D}) \ ,$$

which implies that all the PnP-PGM iterates $\{\boldsymbol{x}^t\}$ are in $\mathsf{Im}(\mathsf{D})$.

Note also the following equivalences

$$\mathsf{Zer}(\nabla g) = \mathsf{Fix}(\mathsf{I} - \gamma \nabla g) = \{\boldsymbol{x} \in \mathbb{R}^n : \nabla g(\boldsymbol{x}) = \boldsymbol{0}\} = \{\boldsymbol{x} \in \mathbb{R}^n : \boldsymbol{A}\boldsymbol{x} = \boldsymbol{y}\} \tag{2a}$$
$$\mathsf{Zer}(\mathsf{R}) = \mathsf{Fix}(\mathsf{D}) = \{\boldsymbol{x} \in \mathbb{R}^n : \mathsf{R}(\boldsymbol{x}) = \boldsymbol{0}\} \ , \tag{2b}$$

where the first equality in (2a) is due to the following equivalence true for any $\boldsymbol{x} \in \mathbb{R}^n$ and $\gamma > 0$

$$\nabla g(\boldsymbol{x}) = \boldsymbol{0} \quad \Leftrightarrow \quad \boldsymbol{x} - \gamma \nabla g(\boldsymbol{x}) = \boldsymbol{x} \ .$$

From the assumption $\boldsymbol{y} = \boldsymbol{A}\boldsymbol{x}^*$ with $\boldsymbol{x}^* \in \mathsf{Zer}(\mathsf{R})$ and from (2), we have the following inclusions:

$$\boldsymbol{x}^* \in \mathsf{Zer}(\nabla g) \cap \mathsf{Zer}(\mathsf{R}) \subseteq \mathsf{Fix}(\mathsf{T}) \subseteq \mathsf{Im}(\mathsf{D}) \subseteq \mathbb{R}^n \ .$$

From Lemma 3 and Lemma 6, we conclude that for any $\boldsymbol{x}, \boldsymbol{z} \in \mathsf{Im}(\mathsf{D})$, we have

$$\|\mathsf{T}(\boldsymbol{x}) - \mathsf{T}(\boldsymbol{z})\|_2 \leq c\|\boldsymbol{x} - \boldsymbol{z}\|_2 \quad \text{with} \quad c = (1 + \alpha)\max\{|1 - \gamma\mu|, |1 - \gamma\lambda|\} \ .$$

From $\mathsf{T}$ being a contraction over $\mathsf{Im}(\mathsf{D})$ and with Lemma 4, we can conclude that $\boldsymbol{x}^* \in \mathsf{Zer}(\nabla g) \cap \mathsf{Zer}(\mathsf{R})$ is the unique fixed point of PnP-PGM for any $\boldsymbol{x}^0 \in \mathsf{Im}(\mathsf{D})$. Thus, we have that

$$\|\boldsymbol{x}^t - \boldsymbol{x}^*\|_2 = \|\mathsf{T}(\boldsymbol{x}^{t-1}) - \mathsf{T}(\boldsymbol{x}^*)\|_2 \leq c\|\boldsymbol{x}^{t-1} - \boldsymbol{x}^*\|_2 \leq \cdots \leq c^t\|\boldsymbol{x}^0 - \boldsymbol{x}^*\|_2 \ ,$$

which establishes the desired result.

## B  Proof of Theorem 2

In this section, we extend the analysis in Section A to the noisy measurement model $\boldsymbol{y} = \boldsymbol{A}\boldsymbol{x}^* + \boldsymbol{e}$ where $\boldsymbol{x}^* \in \mathbb{R}^n$ and $\boldsymbol{e} \in \mathbb{R}^m$. The following analysis builds on that of CSGM in [3] by showing that the proof techniques used for CSGM can be also used for analyzing PnP. Note also that one can improve the error term in the recovery under an additional assumption discussed in Section B.1.

**Theorem 2.** *Run PnP-PGM for $t \geq 1$ iterations under Assumptions 1-2 for the problem (1) of the main paper with $\boldsymbol{x}^* \in \mathbb{R}^n$ and $\boldsymbol{e} \in \mathbb{R}^m$. Then, the sequence $\boldsymbol{x}^t$ generated by PnP-PGM satisfies*

$$\|\boldsymbol{x}^t - \boldsymbol{x}^*\|_2 \leq c\|\boldsymbol{x}^{t-1} - \boldsymbol{x}^*\|_2 + \varepsilon \leq c^t\|\boldsymbol{x}^0 - \boldsymbol{x}^*\|_2 + \frac{\varepsilon(1 - c^t)}{(1 - c)} \ , \tag{3}$$

*where $\boldsymbol{x}^0 \in \mathsf{Im}(\mathsf{D})$ and*

$$\varepsilon := (1 + c)\left[\left(1 + 2\sqrt{\lambda/\mu}\right)\|\boldsymbol{x}^* - \mathsf{proj}_{\mathsf{Zer}(\mathsf{R})}(\boldsymbol{x}^*)\|_2 + 2/\sqrt{\mu}\|\boldsymbol{e}\|_2 + \delta(1 + 1/\alpha)\right] \tag{4}$$

*and $c := (1 + \alpha)\max\{|1 - \gamma\mu|, |1 - \gamma\lambda|\}$ with $\lambda := \lambda_{\max}(\boldsymbol{A}^\mathsf{T}\boldsymbol{A})$.*

Suppose all the assumptions for Theorem 2 are true and the step size $\gamma > 0$ is selected in a way that satisfies eq. (10) of the main manuscript. First note that Lemma 3 and Lemma 6 imply that for $\overline{\boldsymbol{x}} \in \mathsf{Fix}(\mathsf{T})$, we have that

$$\|\boldsymbol{x}^t - \overline{\boldsymbol{x}}\|_2 \leq c\|\boldsymbol{x}^{t-1} - \overline{\boldsymbol{x}}\|_2 \quad \text{with} \quad c = (1 + \alpha)\max\{|1 - \gamma\mu|, |1 - \gamma\lambda|\} \in (0, 1) \ . \tag{5}$$

Let $\widehat{\boldsymbol{x}} = \mathsf{proj}_{\mathsf{Zer}(\mathsf{R})}(\boldsymbol{x}^*)$, then we have that

$$\|\overline{\boldsymbol{x}} - \widehat{\boldsymbol{x}}\| \leq \frac{1}{\sqrt{\mu}}\left[\|\boldsymbol{y} - \boldsymbol{A}\overline{\boldsymbol{x}}\|_2 + \|\boldsymbol{y} - \boldsymbol{A}\widehat{\boldsymbol{x}}\|_2\right]$$

$$\leq \frac{1}{\sqrt{\mu}}\left[\min_{\boldsymbol{x}\in\mathsf{Zer}(\mathsf{R})}\|\boldsymbol{y} - \boldsymbol{A}\boldsymbol{x}\|_2 + \sqrt{\mu}\delta(1 + 1/\alpha) + \|\boldsymbol{y} - \boldsymbol{A}\widehat{\boldsymbol{x}}\|_2\right]$$

$$\leq \frac{2}{\sqrt{\mu}}\|\boldsymbol{y} - \boldsymbol{A}\widehat{\boldsymbol{x}}\|_2 + \delta(1 + 1/\alpha)$$

$$\leq 2\sqrt{\frac{\lambda}{\mu}}\|\boldsymbol{x}^* - \widehat{\boldsymbol{x}}\|_2 + \frac{2}{\sqrt{\mu}}\|\boldsymbol{e}\|_2 + \delta(1 + 1/\alpha)\ ,$$

where the first inequality uses S-REC, the second one uses Lemma 1 in Section B.1, the third one combines two terms by picking the larger one, and the final one uses the measurement model and the triangular inequality. By using the inequality above, we can obtain the bound

$$\|\overline{\boldsymbol{x}} - \boldsymbol{x}^*\|_2 \leq \left[1 + 2\sqrt{\lambda/\mu}\right]\|\boldsymbol{x}^* - \mathsf{prox}_{\mathsf{Zer}(\mathsf{R})}(\boldsymbol{x}^*)\|_2 + [2/\sqrt{\mu}]\|\boldsymbol{e}\|_2 + \delta(1 + 1/\alpha) \coloneqq \varepsilon/(1 + c)\ .$$

Note that the first two terms of $\varepsilon/(1 + c)$ above are the distance of $\boldsymbol{x}^*$ to $\mathsf{Zer}(\mathsf{R})$ and the magnitude of the error $\boldsymbol{e}$, and have direct analogs in standard compressed sensing. The third term is the consequence of the possibility for the solution of PnP not being in $\mathsf{Zer}(\mathsf{R})$ and as discussed in Section B.1 when $\mathsf{Zer}(\mathsf{R}) \cap \mathsf{Zer}(\nabla g) \neq \varnothing$, then the third term disappears.

Then, from (5), we obtain

$$\|\boldsymbol{x}^t - \boldsymbol{x}^*\|_2 \leq \|\boldsymbol{x}^t - \overline{\boldsymbol{x}}\|_2 + \|\overline{\boldsymbol{x}} - \boldsymbol{x}^*\|_2 = \|\boldsymbol{x}^t - \overline{\boldsymbol{x}}\|_2 + \varepsilon/(c + 1)$$

$$\leq c\|\boldsymbol{x}^{t-1} - \overline{\boldsymbol{x}}\|_2 + \varepsilon/(c + 1) = c\|\boldsymbol{x}^{t-1} - \boldsymbol{x}^*\|_2 + c\varepsilon/(c + 1) + \varepsilon/(c + 1)$$

$$= c\|\boldsymbol{x}^{t-1} - \boldsymbol{x}^*\|_2 + \varepsilon \leq c^t\|\boldsymbol{x}^0 - \boldsymbol{x}^*\|_2 + \epsilon\sum_{k=0}^{t-1}c^k$$

$$\leq c^t\|\boldsymbol{x}^0 - \boldsymbol{x}^*\|_2 + \varepsilon(1 - c^t)/(1 - c)\ ,$$

which establishes the desired result.

## B.1 A Technical Lemma for the Proof of Theorem 2

The following lemma provides a bound used for Theorem 2. As discussed within the proof, if $\mathsf{Zer}(\mathsf{R}) \cap \mathsf{Zer}(\nabla g) \neq \varnothing$, the error term on the right of Lemma 1 can be removed by using Lemma 4. While this would lead to a tighter overall bound for Theorem 2, it would also reduce its generality. Fig. 3 empirically shows that the sequence $\|\mathsf{R}(\boldsymbol{x}^t)\|_2$ obtained by PnP-PGM in our simulations converges to a small value, suggesting that the solution obtained by the algorithm is near $\mathsf{Zer}(\mathsf{R})$.

**Lemma 1.** *Under the assumptions of Theorem 2 in the main manuscript, we have*

$$\|\boldsymbol{y} - \boldsymbol{A}\overline{\boldsymbol{x}}\|_2 \leq \min_{\boldsymbol{x}\in\mathsf{Zer}(\mathsf{R})}\|\boldsymbol{y} - \boldsymbol{A}\boldsymbol{x}\|_2 + \sqrt{\mu}\delta(1 + 1/\alpha)\ .$$

*If in addition, we know that* $\mathsf{Zer}(\mathsf{R}) \cap \mathsf{Zer}(\nabla g) \neq \varnothing$, *then*

$$\|\boldsymbol{y} - \boldsymbol{A}\overline{\boldsymbol{x}}\|_2 \leq \min_{\boldsymbol{x}\in\mathsf{Zer}(\mathsf{R})}\|\boldsymbol{y} - \boldsymbol{A}\boldsymbol{x}\|_2\ .$$

*Proof.* First note that by re-expressing the fixed point equation of PnP-PGM, we obtain

$$\overline{\boldsymbol{x}} = \mathsf{D}(\overline{\boldsymbol{x}} - \gamma\nabla g(\overline{\boldsymbol{x}}))$$

$$\Leftrightarrow \quad \begin{cases} \overline{\boldsymbol{z}} = \overline{\boldsymbol{x}} - \gamma\nabla g(\overline{\boldsymbol{x}}) \\ \overline{\boldsymbol{x}} = \overline{\boldsymbol{z}} - (\overline{\boldsymbol{z}} - \mathsf{D}(\overline{\boldsymbol{z}})) = \overline{\boldsymbol{z}} - \mathsf{R}(\overline{\boldsymbol{z}}) \end{cases} \quad \Rightarrow \quad \nabla g(\overline{\boldsymbol{x}}) + \frac{1}{\gamma}\mathsf{R}(\overline{\boldsymbol{z}}) = \boldsymbol{0}\ ,$$

where the final result is obtained by adding the two equalities on the left. Since $g$ satisfies S-REC over $\mathsf{Im}(\mathsf{D})$, Lemma 5 in Section D.2 implies that for any $\boldsymbol{x} \in \mathsf{Im}(\mathsf{D})$ and $\overline{\boldsymbol{x}} \in \mathsf{Fix}(\mathsf{T})$

$$g(\boldsymbol{x}) \geq g(\overline{\boldsymbol{x}}) + \nabla g(\overline{\boldsymbol{x}})^\mathsf{T}(\boldsymbol{x} - \overline{\boldsymbol{x}}) + \frac{\mu}{2}\|\boldsymbol{x} - \overline{\boldsymbol{x}}\|_2^2$$

$$= g(\overline{\boldsymbol{x}}) - (1/\gamma)\mathsf{R}(\overline{\boldsymbol{z}})^\mathsf{T}(\boldsymbol{x} - \overline{\boldsymbol{x}}) + \frac{\mu}{2}\|\boldsymbol{x} - \overline{\boldsymbol{x}}\|_2^2$$

$$\geq \min_{\boldsymbol{x} \in \mathsf{Im}(\mathsf{D})} \left\{ g(\overline{\boldsymbol{x}}) - (1/\gamma)\mathsf{R}(\overline{\boldsymbol{z}})^\mathsf{T}(\boldsymbol{x} - \overline{\boldsymbol{x}}) + \frac{\mu}{2}\|\boldsymbol{x} - \overline{\boldsymbol{x}}\|_2^2 \right\}$$

$$\geq \min_{\boldsymbol{x} \in \mathbb{R}^n} \left\{ g(\overline{\boldsymbol{x}}) - (1/\gamma)\mathsf{R}(\overline{\boldsymbol{z}})^\mathsf{T}(\boldsymbol{x} - \overline{\boldsymbol{x}}) + \frac{\mu}{2}\|\boldsymbol{x} - \overline{\boldsymbol{x}}\|_2^2 \right\}$$

$$\geq g(\overline{\boldsymbol{x}}) - \frac{1}{2\mu\gamma^2}\|\mathsf{R}(\overline{\boldsymbol{z}})\|_2^2 \,,$$

where $\overline{\boldsymbol{z}} = \overline{\boldsymbol{x}} - \gamma\nabla g(\overline{\boldsymbol{x}})$. By rearranging the terms and minimizing over $\boldsymbol{x} \in \mathsf{Im}(\mathsf{D})$, we obtain

$$g(\overline{\boldsymbol{x}}) \leq \min_{\boldsymbol{x} \in \mathsf{Zer}(\mathsf{R})} g(\boldsymbol{x}) + \frac{1}{2\mu\gamma^2}\|\mathsf{R}(\overline{\boldsymbol{z}})\|_2^2 \leq \min_{\boldsymbol{x} \in \mathsf{Zer}(\mathsf{R})} g(\boldsymbol{x}) + \frac{\delta^2}{2\mu\gamma^2} \,, \tag{6}$$

where in the last inequality we used the boundedness of $\mathsf{R}$.

By using the actual expression of $g$ and the lower-bound on $\gamma$ in eq. (10) in the main paper, we obtain $1/\gamma < \mu\,(1 + 1/\alpha)$

$$\Rightarrow \quad \|\boldsymbol{y} - \boldsymbol{A}\overline{\boldsymbol{x}}\|_2 \leq \min_{\boldsymbol{x} \in \mathsf{Zer}(\mathsf{R})} \|\boldsymbol{y} - \boldsymbol{A}\boldsymbol{x}\|_2 + \delta/(\sqrt{\mu}\gamma) \leq \min_{\boldsymbol{x} \in \mathsf{Zer}(\mathsf{R})} \|\boldsymbol{y} - \boldsymbol{A}\boldsymbol{x}\|_2 + \delta\sqrt{\mu}(1 + 1/\alpha) \,.$$

If we assume that $\mathsf{Zer}(\mathsf{R}) \cap \mathsf{Zer}(\nabla g) \neq \varnothing$, then from Lemma 4, we have $\overline{\boldsymbol{x}} \in \mathsf{Zer}(\mathsf{R}) \cap \mathsf{Zer}(\nabla g)$, which implies that $\overline{\boldsymbol{x}} = \overline{\boldsymbol{z}}$ and $\mathsf{R}(\overline{\boldsymbol{z}}) = \mathsf{R}(\overline{\boldsymbol{x}}) = \boldsymbol{0}$. In this case, we can eliminate the error term in (6)

$$g(\overline{\boldsymbol{x}}) \leq \min_{\boldsymbol{x} \in \mathsf{Zer}(\mathsf{R})} g(\boldsymbol{x}) \,.$$

$\square$

## C  Proof of Theorem 3

**Theorem 3.** *Suppose that Assumptions 1-3 are satisfied and that $\mathsf{Zer}(\nabla g) \cap \mathsf{Zer}(\mathsf{R}) \neq \varnothing$, then PnP and RED have the same set of solutions:* $\mathsf{Fix}(\mathsf{T}) = \mathsf{Zer}(\mathsf{G})$.

The SD-RED algorithm in eq. (6) of the main manuscript seeks zeroes of the operator

$$\mathsf{G} = \nabla g + \tau\mathsf{R} \,.$$

It is clear that

$$\nabla g(\boldsymbol{z}) = \boldsymbol{0} \quad \text{and} \quad \mathsf{R}(\boldsymbol{z}) = \boldsymbol{0} \quad \Rightarrow \quad \mathsf{G}(\boldsymbol{z}) = \boldsymbol{0} \,,$$

which corresponds to the inclusion $\mathsf{Zer}(\nabla g) \cap \mathsf{Zer}(\mathsf{R}) \subseteq \mathsf{Zer}(\mathsf{G})$.

We now prove the reverse inclusion under the assumptions of Theorem 3. Let $\boldsymbol{x} \in \mathsf{Zer}(\mathsf{G})$ and $\boldsymbol{z} \in \mathsf{Zer}(\nabla g) \cap \mathsf{Zer}(\mathsf{R})$. Since $\nabla g$ is $\lambda$-Lipschitz continuous with $\lambda = \lambda_{\max}(\boldsymbol{A}^\mathsf{T}\boldsymbol{A})$, Lemma 7 in Section D.2 implies that $\nabla g$ is also $(1/\lambda)$-cocoercive. Therefore, we have that

$$\nabla g(\boldsymbol{x})^\mathsf{T}(\boldsymbol{x} - \boldsymbol{z}) = (\nabla g(\boldsymbol{x}) - \nabla g(\boldsymbol{z}))^\mathsf{T}(\boldsymbol{x} - \boldsymbol{z}) \geq (1/\lambda)\|\nabla g(\boldsymbol{x}) - \nabla g(\boldsymbol{z})\|_2^2 = (1/\lambda)\|\nabla g(\boldsymbol{x})\|_2^2 \,.$$

On the other hand, since $\mathsf{D}$ is nonexpansive, $\mathsf{R} = \mathsf{I} - \mathsf{D}$ is $(1/2)$-cocoercive, which implies that

$$\mathsf{R}(\boldsymbol{x})^\mathsf{T}(\boldsymbol{x} - \boldsymbol{z}) = (\mathsf{R}(\boldsymbol{x}) - \mathsf{R}(\boldsymbol{z}))^\mathsf{T}(\boldsymbol{x} - \boldsymbol{z}) \geq (1/2)\|\mathsf{R}(\boldsymbol{x}) - \mathsf{R}(\boldsymbol{z})\|_2^2 = (1/2)\|\mathsf{R}(\boldsymbol{x})\|_2^2 \,.$$

By using the fact that $\mathsf{G}(\boldsymbol{x}) = \boldsymbol{0}$ and the two inequalities above, we obtain

$$0 = \mathsf{G}(\boldsymbol{x})^\mathsf{T}(\boldsymbol{x} - \boldsymbol{z}) = \nabla g(\boldsymbol{x})^\mathsf{T}(\boldsymbol{x} - \boldsymbol{z}) + \tau\mathsf{R}(\boldsymbol{x})^\mathsf{T}(\boldsymbol{x} - \boldsymbol{z}) \geq (1/\lambda)\|\nabla g(\boldsymbol{x})\|_2^2 + (1/2)\|\mathsf{R}(\boldsymbol{x})\|_2^2 \,, \tag{7}$$

which directly leads to the conclusion

$$\mathsf{G}(\boldsymbol{x}) = \boldsymbol{0} \quad \Rightarrow \quad \nabla g(\boldsymbol{x}) = \boldsymbol{0} \quad \text{and} \quad \mathsf{R}(\boldsymbol{x}) = \boldsymbol{0} \,.$$

Therefore, we have that $\mathsf{Zer}(\mathsf{G}) = \mathsf{Zer}(\nabla g) \cap \mathsf{Zer}(\mathsf{R})$.

Note also that from Lemma 3, we know that when $\mathsf{Zer}(\nabla g) \cap \mathsf{Zer}(\mathsf{R}) \neq \varnothing$, we have $\mathsf{Fix}(\mathsf{T}) = \mathsf{Zer}(\nabla g) \cap \mathsf{Zer}(\mathsf{R})$, which directly leads to our result

$$\mathsf{Zer}(\mathsf{G}) = \mathsf{Zer}(\nabla g) \cap \mathsf{Zer}(\mathsf{R}) = \mathsf{Fix}(\mathsf{T}) \,.$$

# D Background Material

The results in this sections are well-known and can be found in different forms in standard textbooks [1, 5–7]. For completeness, we summarize the key results used in our analysis.

## D.1 Properties of Monotone Operators

**Definition 1.** *An operator $\mathsf{T}$ is Lipschitz continuous with constant $\lambda > 0$ if*

$$\|\mathsf{T}(\boldsymbol{x}) - \mathsf{T}(\boldsymbol{z})\|_2 \leq \lambda \|\boldsymbol{x} - \boldsymbol{z}\|_2 \quad \forall \boldsymbol{x}, \boldsymbol{y} \in \mathbb{R}^n .$$

*When $\lambda = 1$, we say that $\mathsf{T}$ is nonexpansive. When $\lambda < 1$, we say that $\mathsf{T}$ is a contraction.*

**Definition 2.** $\mathsf{T}$ *is monotone if*

$$(\mathsf{T}(\boldsymbol{x}) - \mathsf{T}(\boldsymbol{z}))^{\mathsf{T}}(\boldsymbol{x} - \boldsymbol{z}) \geq 0 \quad \forall \boldsymbol{x}, \boldsymbol{y} \in \mathbb{R}^n .$$

*We say that $\mathsf{T}$ is strongly monotone with parameter $\theta > 0$ if*

$$(\mathsf{T}(\boldsymbol{x}) - \mathsf{T}(\boldsymbol{z}))^{\mathsf{T}}(\boldsymbol{x} - \boldsymbol{z}) \geq \theta \|\boldsymbol{x} - \boldsymbol{z}\|_2^2 \quad \forall \boldsymbol{x}, \boldsymbol{y} \in \mathbb{R}^n .$$

**Definition 3.** $\mathsf{T}$ *is cocoercive with constant $\beta > 0$ if*

$$(\mathsf{T}(\boldsymbol{x}) - \mathsf{T}(\boldsymbol{z}))^{\mathsf{T}}(\boldsymbol{x} - \boldsymbol{z}) \geq \beta \|\mathsf{T}(\boldsymbol{x}) - \mathsf{T}(\boldsymbol{z})\|_2^2 \quad \forall \boldsymbol{x}, \boldsymbol{z} \in \mathbb{R}^n .$$

*When $\beta = 1$, we say that $\mathsf{T}$ is firmly nonexpansive.*

**Definition 4.** *For a constant $0 < \alpha < 1$, we say that $\mathsf{T}$ is $\alpha$-averaged, if there exists a nonexpansive operator $\mathsf{N}$ such that $\mathsf{T} = (1 - \alpha)\mathsf{I} + \alpha\mathsf{N}$.*

The following lemma is derived from the definitions above.

**Lemma 2.** *Consider $\mathsf{R} = \mathsf{I} - \mathsf{D}$ where $\mathsf{D} : \mathbb{R}^n \rightarrow \mathbb{R}^n$. Then,*

$$\mathsf{D} \textit{ is nonexpanisve} \quad \Leftrightarrow \quad \mathsf{R} \textit{ is } (1/2)\textit{-cocoercive} .$$

*Proof.* First suppose that $\mathsf{R}$ is $1/2$ cocoercive. Let $\boldsymbol{h} := \boldsymbol{x} - \boldsymbol{z}$ for any $\boldsymbol{x}, \boldsymbol{z} \in \mathbb{R}^n$. We then have

$$\frac{1}{2}\|\mathsf{R}(\boldsymbol{x}) - \mathsf{R}(\boldsymbol{z})\|_2^2 \leq (\mathsf{R}(\boldsymbol{x}) - \mathsf{R}(\boldsymbol{z}))^{\mathsf{T}}\boldsymbol{h} = \|\boldsymbol{h}\|_2^2 - (\mathsf{D}(\boldsymbol{x}) - \mathsf{D}(\boldsymbol{z}))^{\mathsf{T}}\boldsymbol{h} .$$

We also have that

$$\frac{1}{2}\|\mathsf{R}(\boldsymbol{x}) - \mathsf{R}(\boldsymbol{z})\|_2^2 = \frac{1}{2}\|\boldsymbol{h}\|^2 - (\mathsf{D}(\boldsymbol{x}) - \mathsf{D}(\boldsymbol{z}))^{\mathsf{T}}\boldsymbol{h} + \frac{1}{2}\|\mathsf{D}(\boldsymbol{x}) - \mathsf{D}(\boldsymbol{z})\|_2^2 .$$

By combining these two and simplifying the expression

$$\|\mathsf{D}(\boldsymbol{x}) - \mathsf{D}(\boldsymbol{z})\|_2 \leq \|\boldsymbol{h}\|_2 .$$

The converse can be proved by following this logic in reverse. $\square$

The following lemma relates the Lipschitz continuity of the residual $\mathsf{S} = \mathsf{I} - \mathsf{T}$ to that of $\mathsf{T}$.

**Lemma 3.** *The operator $\mathsf{R} = \mathsf{I} - \mathsf{D}$ is $\alpha$-Lipschitz continuous if and only if the operator $(1/(1+\alpha))\mathsf{D}$ is nonexpansive and $\alpha/(1 + \alpha)$-averaged.*

*Proof.* See Lemma 9 in [4]. $\square$

The following lemma considers fixed points of a composite operator.

**Lemma 4.** *Let $\mathsf{T} = \mathsf{D} \cdot \mathsf{S}$ with $\mathsf{Fix}(\mathsf{D}) \cap \mathsf{Fix}(\mathsf{S}) \neq \varnothing$ be a contraction with constant $\lambda \in (0, 1)$ over the set $\mathsf{Im}(\mathsf{D}) \subseteq \mathbb{R}^n$. Then, we have that $\mathsf{Fix}(\mathsf{T}) = \mathsf{Fix}(\mathsf{D}) \cap \mathsf{Fix}(\mathsf{S})$ .*

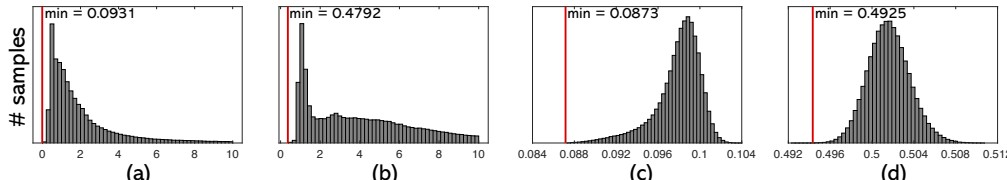

Figure 2: *Empirical evaluation of the S-REC constant* $\mu > 0$ *for the measurement operators* $\boldsymbol{A}$ *used in our simulations. We tested both the AWGN denoisers and the AR operators by randomly sampling from their image spaces* $\mathsf{Im(D)}$. *The* $x$−*axis is the value of* $\|\boldsymbol{Ax} - \boldsymbol{Ay}\|_2^2 / \|\boldsymbol{x} - \boldsymbol{y}\|_2^2$. *(a) and (b) show the histograms for the radially sub-sampled MRI matrices at* $10\%$ *and* $50\%$ *sampling ratios, respectively. (c) and (d) show the histograms for the random Gaussian matrices for the same two sampling ratios. As expected, one can observe the increase in* $\mu$ *for the higher sampling ratio of* $50\%$.

*Proof.* We modify the proof of Proposition 4.49 from [1] to be consistent with our assumptions.

It is clear that $\mathsf{Fix(D)} \cap \mathsf{Fix(S)} \subseteq \mathsf{Fix(T)}$ and our goal is to show the reverse inclusion. Let $\boldsymbol{x} \in \mathsf{Fix(T)}$ and consider three cases.

- *Case* $\mathsf{S(x)} \in \mathsf{Fix(D)}$: We have that
$$\mathsf{S(x)} = \mathsf{D(S(x))} = \mathsf{T(x)} = \boldsymbol{x} \in \mathsf{Fix(D)} \cap \mathsf{Fix(S)} \ .$$

- *Case* $\boldsymbol{x} \in \mathsf{Fix(S)}$: We have that
$$\mathsf{D(x)} = \mathsf{D(S(x))} = \mathsf{T(x)} = \boldsymbol{x} \in \mathsf{Fix(D)} \cap \mathsf{Fix(S)} \ .$$

- *Case* $\mathsf{S(x)} \notin \mathsf{Fix(D)}$ *and* $\boldsymbol{x} \notin \mathsf{Fix(S)}$: Since $\mathsf{T} = \mathsf{D} \cdot \mathsf{S}$ is a contraction over $\mathsf{Im(D)}$
$$\|\boldsymbol{x} - \boldsymbol{z}\|_2 = \|\mathsf{T(x)} - \mathsf{T(z)}\|_2 \leq \lambda \|\boldsymbol{x} - \boldsymbol{z}\|_2 \quad \forall \boldsymbol{z} \in \mathsf{Fix(D)} \cap \mathsf{Fix(S)} \ ,$$

  which is impossible.

$\square$

## D.2 Convexity, restricted strong convexity, and set-restricted eigenvalue condition

S-REC in the main manuscript can be generalized to the *restricted strong convexity (RSC)* assumption, which is widely-used in the nonconvex analysis of the gradient methods (see Section 3.2 in [8]).

**Definition 5.** *A continuously differentiable function* $g$ *is said to satisfy restricted strong convexity (RSC) over* $\mathcal{X} \subseteq \mathbb{R}^n$ *with* $\mu > 0$ *if*
$$g(\boldsymbol{z}) \geq g(\boldsymbol{x}) + \nabla g(\boldsymbol{x})^\mathsf{T}(\boldsymbol{z} - \boldsymbol{x}) + \frac{\mu}{2}\|\boldsymbol{z} - \boldsymbol{x}\|_2^2 \quad \forall \boldsymbol{x}, \boldsymbol{z} \in \mathcal{X} \ .$$

In fact, for $g(\boldsymbol{x}) = \frac{1}{2}\|\boldsymbol{y} - \boldsymbol{Ax}\|_2^2$, S-REC is equivalent to RSC in Definition 5.

**Lemma 5.** *Let* $g(\boldsymbol{x}) = \frac{1}{2}\|\boldsymbol{y} - \boldsymbol{Ax}\|_2^2$ *and consider* $\mathcal{X} \subseteq \mathbb{R}^n$. *Then,*

$$g \text{ satisfies S-REC with } \mu \text{ over } \mathcal{X} \quad \Leftrightarrow \quad g \text{ satisfies } \mu\text{-RSC over } \mathcal{X} \ .$$

*Proof.* Suppose $g$ is the least-squares function that satisfies S-REC with $\mu$, then for any $\boldsymbol{x}, \boldsymbol{z} \in \mathcal{X}$
$$
\begin{aligned}
g(\boldsymbol{z}) &= g(\boldsymbol{x}) + \nabla g(\boldsymbol{x})^\mathsf{T}(\boldsymbol{z} - \boldsymbol{x}) + \frac{1}{2}(\boldsymbol{z} - \boldsymbol{x})\boldsymbol{A}^\mathsf{T}\boldsymbol{A}(\boldsymbol{z} - \boldsymbol{x}) \\
&= g(\boldsymbol{x}) + \nabla g(\boldsymbol{x})^\mathsf{T}(\boldsymbol{z} - \boldsymbol{x}) + \frac{1}{2}\|\boldsymbol{A}(\boldsymbol{z} - \boldsymbol{x})\|_2^2 \\
&\geq g(\boldsymbol{x}) + \nabla g(\boldsymbol{x})^\mathsf{T}(\boldsymbol{z} - \boldsymbol{x}) + \frac{\mu}{2}\|\boldsymbol{z} - \boldsymbol{x}\|_2^2 \ ,
\end{aligned}
$$

which implies that $g$ satisfies $\mu$-RSC. To show S-REC using $\mu$-RSC, follow the logic in reverse. $\square$

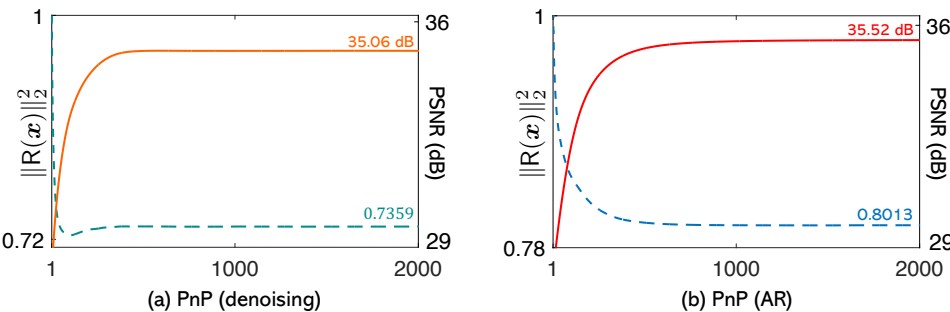

Figure 3: *Illustration of the convergence of PnP under nonexpensive denoisers and AR operators. Average normalized distance to $\|\mathsf{R}(\boldsymbol{x})\|_2^2 = \|\boldsymbol{x} - \mathsf{D}(\boldsymbol{x})\|_2^2$ and PSNR (dB) are plotted as dashed and solid lines, respectively, against the iteration number. This plot illustrates that PnP in our experiments converges to vectors close to $\mathsf{Zer}(\mathsf{R})$, which is consistent with the view that it regularizes inverse problems by obtaining solutions near the fixed-points of a denoiser/AR operator.*

One can use the previous and the following lemma to show that the gradient step of PnP-PGM can be a contraction for any vector in $\mathsf{Im}(\mathsf{D})$ for a properly chosen step size.

**Lemma 6.** *Assume $g$ satisfies $\mu$-RSC over $\mathcal{X} \subseteq \mathbb{R}^n$ and $\nabla g$ is $\lambda$-Lipschitz continuous. Then,*

$$\|(\mathsf{I} - \gamma \nabla g)(\boldsymbol{x}) - (\mathsf{I} - \gamma \nabla g)(\boldsymbol{z})\|_2 \leq \max\{|1 - \gamma\mu|, |1 - \gamma\lambda|\}\|\boldsymbol{x} - \boldsymbol{z}\|_2 \quad \forall \boldsymbol{x}, \boldsymbol{z} \in \mathcal{X} .$$

*Proof.* Since for any $\boldsymbol{x}, \boldsymbol{z} \in \mathcal{X}$, the function $g$ is strongly convex with constant $\mu$, this lemma is a simple modification of Lemma 7 in [4]. □

**Lemma 7.** *For a convex and continuously differentiable function $g$, we have*

$$\nabla g \text{ is } \lambda\text{-Lipschitz continuous} \quad \Leftrightarrow \quad \nabla g \text{ is } (1/\lambda)\text{-cocoercive} .$$

*Proof.* See Theorem 2.1.5 in Section 2.1 of [7]. □

## E   Additional Technical Details and Numerical Results

We designed two types of deep priors for PnP/RED: (i) an AWGN denoiser and (ii) an artifact-removal (AR) operator trained to remove artifacts specific to the PnP iterations[1]. Both of these deep priors share the same neural network architecture, based on DnCNN [9]. The networks contain three components. The first part is a composite convolutional layer, consisting of a normal convolutional layer and a rectified linear units (ReLU) layer. It convolves the $n_1 \times n_2$ input to $n_1 \times n_2 \times 64$ features maps by using 64 filters of size $3 \times 3$. The second part is a sequence of 10 composite convolutional layers, each having 64 filters of size $3 \times 3 \times 64$. Those composite layers further process the feature maps generated by the first part. The third part of the network, a single convolutional layer, generates the final output image by convolving the feature maps with a $3 \times 3 \times 64$ filter. Every convolution is performed with a stride $= 1$, so that the intermediate feature maps share the same spatial size of the input image. We train several denoisers to optimize the *mean squared error (MSE)* by using the Adam optimizer. All the experiments in this work were performed on a machine equipped with an Intel Xeon Gold 6130 Processor and eight NVIDIA GeForce RTX 2080 Ti GPUs.

We now present the implementation details of training the AR operators used in this work. Inspired by ISTA-Net[+] [2], we implement our own deep unfolding neural network for training the AR operator. Given an initial solution $\boldsymbol{x}^0$, *i.e.* $\boldsymbol{x}^0 = \boldsymbol{A}^\mathsf{T}\boldsymbol{y}$, we iteratively refine it by infusing information from both the gradient of the data-fidelity term $\nabla g$ and the learned operator $\mathsf{D}$ defined as

$$\mathsf{R}(\boldsymbol{x}) = (\mathsf{I} - \mathsf{D})(\boldsymbol{x}) = \boldsymbol{x} - \mathsf{D}(\boldsymbol{x}) , \tag{8}$$

---

[1]The implementation of our pre-trained denoisers and AR operators are also available in the supplement.
[2]ISTA-Net[+] is publicly available at https://github.com/jianzhangcs/ISTA-Net-PyTorch.

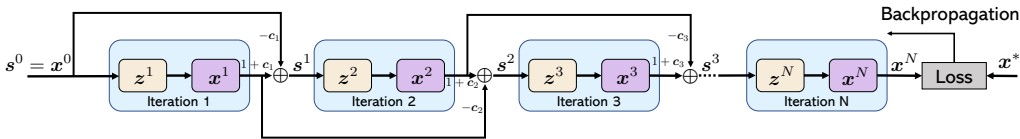

Figure 4: *Detailed architecture used for training the AR operator by unrolling the iterations of PnP-FISTA [10] with the DnCNN prior. Each layer contains one iteration consisting of a data-consistency update and an image prior update. The input of the unrolling network is the initialization $x^0$ and the output is the reconstructed image from the $N$th iteration, which is subsequently used within the training loss. In order to make the AR operator satisfy the Assumption A, we impose the spectral normalization and weight sharing on DnCNN across different iterations. Note that once DnCNN is pre-trained following this scheme, it is used as an AR operator within PnP.*

where R is the residual of the deep neural network. We use Nesterov acceleration in the unrolled architecture, fixing the total number of unrolling iterations to $N \geq 1$

$$z^k = s^{k-1} - \gamma \nabla g(s^{k-1}) \tag{9}$$

$$x^k = \mathsf{D}(z^k) \tag{10}$$

$$c_k = (q_{k-1} - 1)/q_k \tag{11}$$

$$s^k = x^k + c_k(x^k - x^{k-1}) \, , \tag{12}$$

where $\gamma > 0$ is a step-size parameter and the value of $q_k = 1/2(1 + \sqrt{1 + 4q_{k-1}^2})$ is adapted during the training for better PSNR performance. Fig. 4 illustrates the algorithmic details for training the AR operator. In our implementation, we opted to share the weights of the AR operator across different iterations to satisfy our theoretical assumptions. We trained several AR operators for $N$ unfolded iterations using the MSE loss

$$\mathcal{L}_{\textbf{MSE}} = \frac{1}{M} \sum_{j=1}^{M} \|x_j^N - x_j^*\|_2^2 \, , \tag{13}$$

where $x_j^*$ is the ground truth. We also included a *smoothness-constraint* loss across different iterations, defined as

$$\mathcal{L}_{\textbf{Smooth}} = \frac{1}{M} \sum_{j=1}^{M} \sum_{k=N-q}^{N} \|x_j^k - \mathsf{D}(z_j^k)\|_2^2 \, . \tag{14}$$

We observe that the AR operators trained with this smoothness-constraint outperform those trained without it, especially, when the AR operator is integrated into the PnP algorithm. The total AR training loss is thus $\mathcal{L} = \mathcal{L}_{\textbf{MSE}} + \beta \mathcal{L}_{\textbf{Smooth}}$, where $\beta > 0$ controls the amount of smoothing. For the experiments in this paper, we set $N = 90, \beta = 10$ for all gray and color AR operators' training, while we set $N = 27, \beta = 10$ for CS-MRI.

We used a pre-training strategy to accelerate the training of the weights within the AR operator. Since the weights are shared across iterations of the deep unfolding network, we can then initialize them with those obtained from pre-trained AWGN denoisers. We observe that this pre-training strategy is considerably more efficient than initializing the entire unfolding network with the random weights. Since we initialize our learned components with deep denoisers, the initial setup for our method exactly corresponds to tuning a PnP approach with a deep denoiser. Such training adapts the operator D to a particular inverse problem and data distribution.

In Fig. 2, we report the empirical evaluation of $\mu$ for the measurement operators used in our experiments by sampling images from $\mathsf{Im}(\mathsf{D})$. Specifically, we test two types of measurement matrixes for CS, namely random matrix and radially subsampled Fourier matrix, both at subsampling rates of $10\%$ and $50\%$. For each type of matrix, we first use the operator D to generate several denoised image pairs on BSD68 and medical brain images, respectively. This ensures the tested image pairs are all in the range of D. We plot the histograms of $\mu = \|Ax - Az\|_2^2/\|x - z\|_2^2$, and the minimum value of each histogram is indicated by a vertical bar, providing an empirical lower bound on the values of $\mu$.

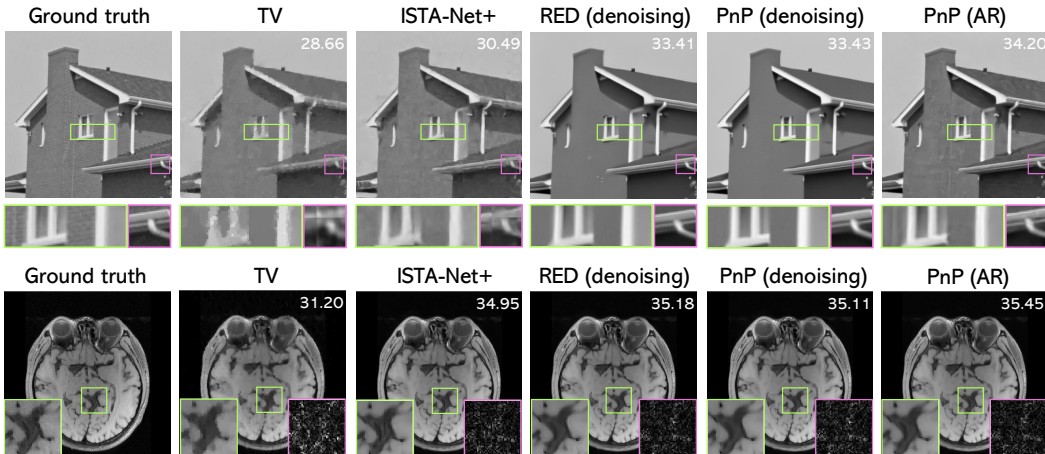

Figure 5: *Additional visual comparisons between various methods for CS and CS-MRI. Top: reconstruction results on the "House" image in Set11 at CS ratios of* 10%. *Bottom: results on MRI images with radially under-sampling at CS ratios of* 20% *(The pink box provides the error residual that was amplified by* 10× *for better visualization.). Best viewed by zooming in the display.*

Table 1: Average PSNR (dB) values for two spectral normalization (SN) technique used in training $\alpha$-Lipschitz continuity denoisers on Set11.

| Method
CS Ratio | PnP (denoiser real-SN [4]) | PnP (denoiser SN [11]) |
|---|---|---|
| 0.1 | 27.32 | 27.76 |
| 0.3 | 34.78 | 35.06 |

Fig. 2 illustrates that empirically the measurement operators $\boldsymbol{A}$ used in this work satisfies S-REC over $\mathsf{Im}(\mathsf{D})$ with $\mu > 0$.

In Table 1, we provide additional empirical comparisons between the spectral normalization (SN) technique in [4] and the one in [11] for training denoisers used in PnP. It is worth noting that the SN from [11] uses a convenient but inexact implementation for the convolutional layers. Both of our pre-trained models are available here: `https://github.com/wustl-cig/pnp-recovery`.

In Fig. 3, we report the convergence of $\|\mathsf{R}(\boldsymbol{x}^t)\|_2^2 = \|\boldsymbol{x}^t - \mathsf{D}(\boldsymbol{x}^t)\|_2^2$ for both the AWGN denoisers and the AR operators use in our experiments. As can be observed from the plots, in both cases, PnP converges to vectors close to $\mathsf{Zer}(\mathsf{R})$, which is consistent with the view that it regularizes inverse problems by obtaining solutions near the fixed-points of a denoiser/AR operator. Note that this view is completely backward compatible with the traditional sparsity-promoting priors and ISTA-algorithms, where one achieves regularization by promoting sparse solutions in some transform domain.

We ran fixed-point iterations of the denoisers and the AR operators used in this work on Set11. Fig. 6 below presents visual comparisons for different values of $\|\mathsf{R}(\boldsymbol{x})\|_2^2$ for the AR operator and denoiser, respectively. Table 2 provides PSNR (dB) for different values of $\|\mathsf{R}\|_2^2$ using TV as a reference. In all experiments, we observed that as the images get closer to the fixed-points of the denoiser, they start losing visual details. On the other hand, deep denoisers seem to preserve visual details better than TV. This suggests that the regularization for the deep priors we used in this work is analogous to that of traditional regularization using TV, where good performance is achieved by finding images balancing data-consistency and the distance to the fixed-points of D but not by returning the fixed-points of D directly. Note that for some denoisers it might be desirable to directly return the images at the fixed points. For example, consider a denoiser that projects vectors to a set of natural images; the fixed-points of such denoiser are natural images.

We provide additional visualizations of the solutions produced by PnP/RED and various baseline methods considered in our work. Fig. 5 (top) reports the visual comparison of multiple methods on Set11 with CS ratios of 10%, while Fig. 5 (bottom) reports the comparison on medical brain

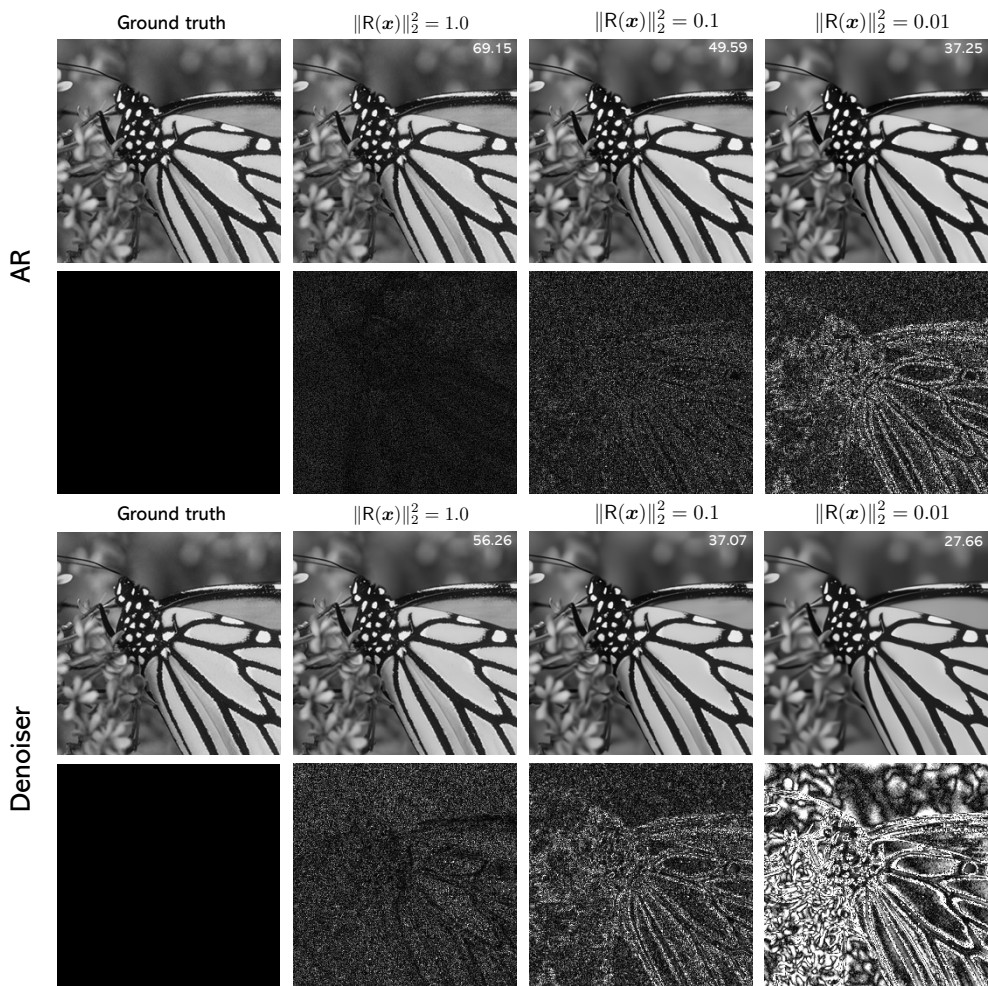

Figure 6: *Visual comparison of running fixed-point iterations of the AR operators and denoisers used in the main paper when applied to the Set11. Table 2 additionally provides PSNR (dB) for different values of $\|\mathsf{R}x\|_2^2$ using TV as a reference. The error residual to the ground truth images was amplified $30\times$ and showed in grayscale for better visualization.*

Table 2: Average PSNR (dB) values for different values of $\|\mathsf{R}(x)\|_2^2$ on Set 11.

| $\|\mathsf{R}(x)\|_2^2$ 
 CS Ratio | 1.0 | 0.10 | 0.001 |
|---|---|---|---|
| **AR** | 72.72 | 43.98 | 35.41 |
| **Denoiser** | 65.76 | 35.37 | 23.81 |
| **TV** | 14.45 | 14.24 | 13.84 |

images for CS-MRI with under-sampling ratios of 20%. Fig. 7 illustrates the numerical comparison on BSD68 for CS ratios of 30% (top) and 10% (bottom), respectively. Fig. 9 reports the visual comparison between PnP/RED and two CS methods based on StyleGAN2. Note that in all figures, PnP/RED achieves competitive results, with PnP (AR) achieving superior reconstruction results compared to PnP (denoising).

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

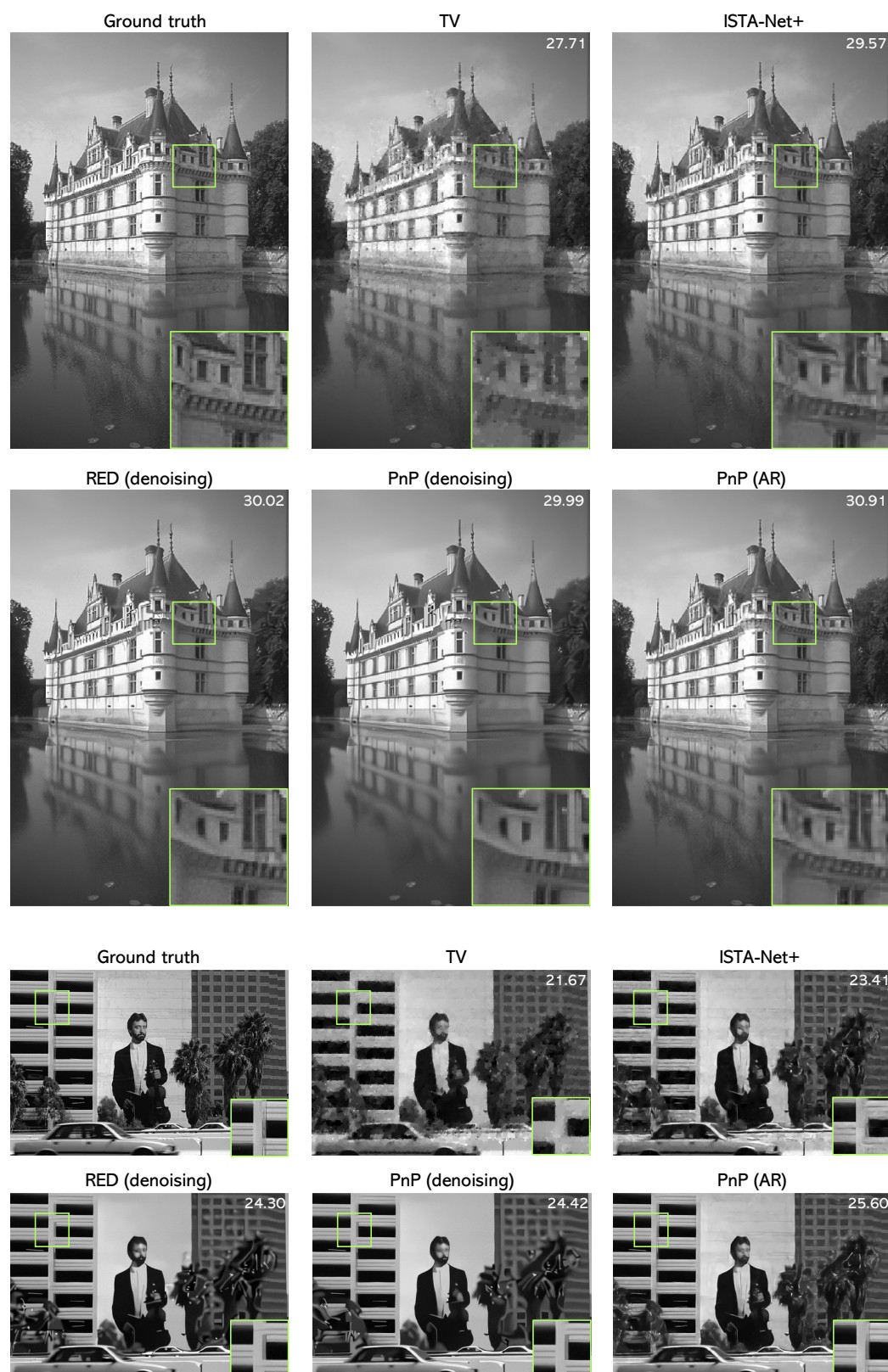

Figure 7: *Supplementary visual comparisons between various methods on BSD68. Top: Results at 30% sampling ratio. Bottom: Results at 10% sampling ratio.*

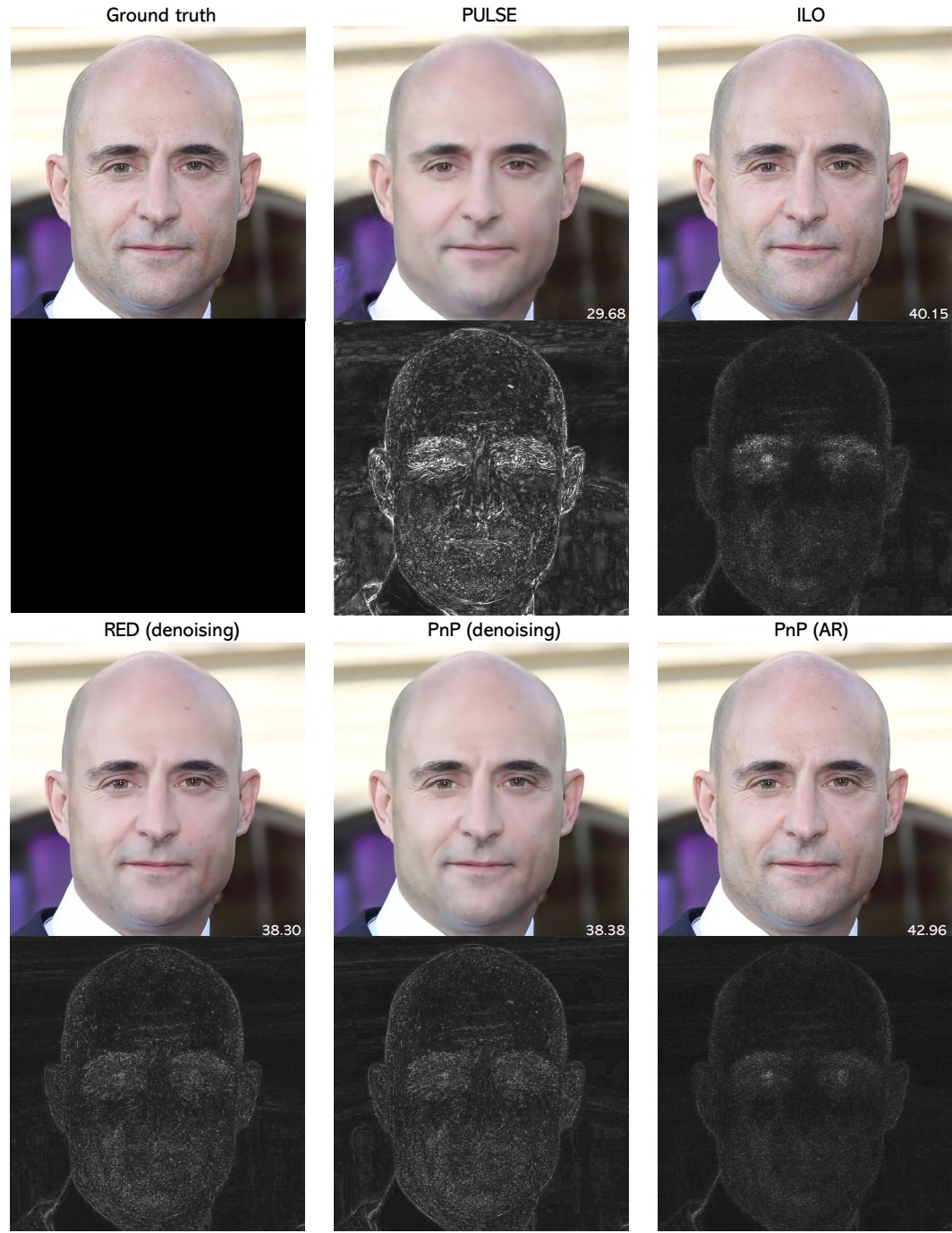

Figure 8: *Visual comparison between PnP/RED and two methods using generative models, when applied to the CelebA HQ dataset at $10\%$ sampling ratio. The error residuals relative to the ground truth images were amplified $10\times$ and showed in grayscale for better visualization. Note the similarity between the RED and PnP solutions. PnP (AR) leads to sharper images, comparable to those obtained by ILO with StyleGAN2. This highlights the benefit of using pre-trained AR operators.*

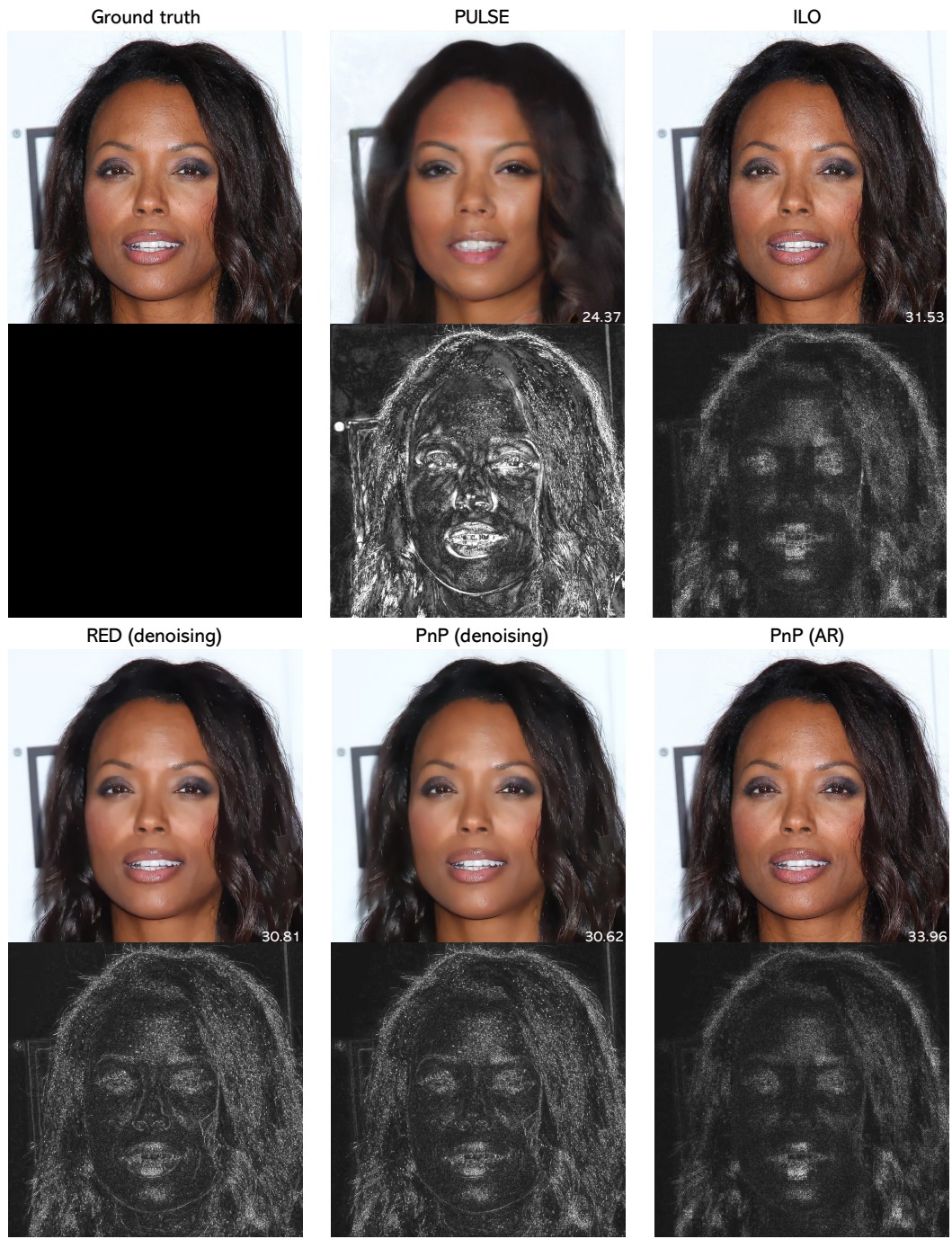

Figure 9: *Visual comparison between PnP/RED and two methods using generative models, when applied to the CelebA HQ dataset at $10\%$ sampling ratio. The error residuals to relative to the ground truth images were amplified $7\times$ and showed in grayscale for better visualization. Note the similarity between the RED and PnP solutions. PnP (AR) leads to sharper images, comparable to those obtained by ILO with StyleGAN2. This highlights the benefit of using pre-trained AR operators.*