# OpenReview forum: "Recovery Analysis for Plug-and-Play Priors using the Restricted Eigenvalue Condition"
_NeurIPS.cc/2021/Conference — NeurIPS 2021 Poster_

### Official Review · Reviewer_4SHo · 2021-07-15

**Rating:** 7
**Confidence:** 4

**Summary:**

This paper contains three theoretical results as well as a numerical comparison for the regularization by denoising (RED) and plug-and-play prior (PnP) algorithms, in which energy minimization algorithms for inverse problems are used as a template to derive algorithmic schemes that utilize denoising (or artifact removal) networks instead of regularizers. The paper shows that under (compressed-sensing-like) assumptions the recovery of certain solutions can be guaranteed (Theorem 1), and subsequently extends it to an error bound in the presence of noise and no assumptions on the solution to be recovered (Theorem 2). Finally, it is shown that RED and PnP have the same fix point sets if the used denoising network is non-expansive (Theorem 3). In the numerical results, a comparison on the exemplary problem of compressed sensing image recovery is conducted to conclude that artifact removal networks (specialized on removing artifacts that arise in the considered reconstruction problem) perform significantly better than generic denoising networks.

**Limitations And Societal Impact:**

I think the limitations and impact are adequately discussed.

**Main Review:**

The paper is very well-written, the motivation is clear, and the theorems are presented in an easy-to-follow way. Thus, the paper fully convinced me in terms of its clarity. Similarly, I do think the work is significant in the sense that getting a better theoretical understanding and control of the well-working approaches that utilize deep learning is important. Considering the success of compressed sensing, I consider this work as a small next step towards establishing similar results for PnP/RED priors. The presented theoretical results are new, but also not surprising. Moreover, the numerical experiments are almost orthogonal to the theory. This is partially due to the difficulty of verifying the assumptions from the theorems (as the authors themselves rightfully mention as a limitation in section 5). Yet, an in-depth discussion on what the conditions of the theorem mean / what kind of guarantees they provide would have been interesting. For instance,
- assuming estimates as they arise in compressed sensing (e.g. assuming Im(D) \subset k-sparse vectors) how Lipschitz continuous (value of \alpha) should D be for c in Theorem 1 to be less than one?
- The estimate in Thm. 2 crucially depends on the distance of $x^*$ to Zer(R). While the supplement shows that the algorithm converges to values of $\|R(x^t)\|$ that are somewhat small, I am not sure how large the contribution of this part of the error term is. I would be curious to see different results of a fixed point iteration of $D$. I would hope for this to converge to a point in Zer(R), and this might illustrate the thin line between desirable properties for convergence (averaged / contractive operators), and performance. In line 226 it is stated that "both R and D are contractive operators", which, however, means Zer(R) is a singleton and the distance term in Theorem 2 has to be large for some images. Please comment on this.

In summary, I believe this is a well-written paper with some new results that could spark further research in the important topic of understanding the behavior when using generic denoising / artifact removal networks as regularizers.

Minor comments
- Regarding Fig. 1, in my experience trying to adversarially find points at which the network indicates a large Lipschitz constant gives a better impression of the true Lipschitz constant than random sampling. In particular, I am not sure if $D$ really is a contraction (while also believing that this is not necessarily desirable as pointed out above).
- Before eq. (10) it says "is less that one if" >> "than"
- In line 206 "the residual R of the AR operator still satisfies Lipschitz continuity assumptions". Does it satisfy a Lipschitz continuity assumption (which is something any conv-relu network does because it is piecewise linear with finitely many pieces), or does is it 1 or 2 Lipschitz (as the denoisers are). Since imposing Lipschitz continuity reduces the denoising performance this is important to state.
- The $c$ in the line before 52 in the supplementary material is missing a power $t$

----------------------
After writing the above review I checked the code in more detail: The "SpectralNorm" class seems to run the power methods on the weights of convolutional layers by considering the weights as a matrix. I don't think that this computes the right normalization constant as the results may differ from computing the power method for actual convolutions or doing the appropriate  computation in Fourier space. I think it is worth checking the difference.  I also tested the Lipschitz assumption, and it can break for small perturbations, e.g. when choosing
u1 = torch.randn([1, 1, 20, 20]).to(device)
u2 = u1+ 1e-5*torch.randn([1, 1, 20, 20]).to(device)
f1 = dncnnModel(u1)
f2 = dncnnModel(u2)
(yielding Lipschitz constants of around 7 for the provided models) but this is likely due to the limited precision / numerical errors?

------------------------
I would like to thank the authors for their detailed explanations in the rebuttal. I think this is a very nice piece of work!

**Time Spent Reviewing:**

5

---

> ### Author Response · Authors · 2021-08-09
> **Response to Reviewer 4SHo**
>
> Thank you for the careful reading of the paper and positive assessment of our work. Below, we provide point-by-point responses to your comments, which will enable us to improve the paper. In order to provide better answers to your comments, we have performed additional simulations and implemented an alternative spectral normalization technique from [23], which was specifically designed for convolutional neural networks.
>
> Main Review:
> - The revised manuscript will clarify that to have $c < 1$, one needs to have $\alpha < 2\mu/(\lambda-\mu)$. Note how $\alpha$ relates to both the S-REC constant ($\mu$) and the largest eigenvalue of the measurement matrix ($\lambda_{\max}$). For example, this can be related to the traditional k-sparse signal recovery by considering the residual of a smoothed hard-thresholding function as the operator $R$. The bound on $\alpha$ above could then be interpreted as a constraint on the derivative of this function.
>
> - Indeed, as you point out, there is a nontrivial relationship between the distance to the fixed-points of $D$ and the quality of the final recovery. An analogous relationship also exists for traditional denoisers such as soft-thresholding and total variation (TV), where the fixed points are the zero-vector and constant images, respectively. In traditional regularization, the best performance is not achieved by returning the fixed-points of the denoiser, but by promoting solutions close to those fixed-points. Prompted by your remark, we ran fixed-point iterations, starting from images from Set11, of the denoisers and the AR operators used in this paper. The table below provides PSNR (dB) against the value of $||R(x)||^2$. In all experiments, we observed that as the images get closer to the fixed-points of the denoiser, they start losing visual details. On the other hand, deep denoisers seem to preserve visual details better than TV.  This suggests that the regularization for the deep priors we used in the paper is analogous to that of traditional regularization using TV, where good performance is achieved by finding images balancing data-consistency and the distance to the fixed-points of $D$, but not by returning the fixed-points of $D$ directly. Finally, note that for some denoisers it might be desirable to directly return the images at the fixed points. For example, consider a denoiser that projects vectors to a set of natural images; the fixed-points of such denoiser are natural images. We will include these numerical results along with visual examples in the revised supplementary material.
>                  ||R(x)||2              1e-0         1e-1          1e-2
>                     TV                 14.45        14.24         13.84
>                   Denoiser             65.76        35.37         23.81
>                     AR                 72.72        43.98         35.41
>
> - Related to the point above, note that the nonexpansiveness of $D$ is only required for Theorem 3 and can be significantly weakened. As can be seen in line 77 of the supplementary material, we do not need $D$ to be nonexpansive everywhere, but only at the fixed point of RED-SD (see definitions of $x$ and $z$ in lines 74 and 75). Similarly, we do not require a contractive $R\\;(\alpha < 1)$ for Theorems 1 and 2, so long as $c < 1$.
>
> Minor Comments:
>
> - Thank you for highlighting the limitations of the spectral normalization (SN) technique from [63] that we used for training our denoisers. After reviewing our implementation, we agree that the SN in [63] uses a convenient but inexact implementation for convolutional layers. Prompted by this observation, we also implemented the real-SN method from the reference [23] in the paper. We will mention both SN strategies in the revised manuscript and will publicly share both of our pre-trained models with our code upon acceptance. Below we provide a table showing the empirical performance of PnP using those two denoisers on Set11.
>
>                   CS ratio     PnP (denoiser real-SN [23])      PnP (denoiser SN [63])
>                     0.1                 27.32                          27.76
>                     0.3                 34.78                          35.06
>
> - We repeated the experiment that you mention in your review (e.g. where |u1-u2| <= 1e-5*torch.randn([1, 1, 20, 20])) for our models and the models pre-trained using real-SN from [23]. All models satisfy the Lipschitz assumption on natural images for small perturbations of 1e-5. When u1 and u2 are random vectors, both models yield Lipschitz constants > 1 for small perturbations, which suggests that this phenomenon is likely due to limited precision/numerical errors. In any case, as we mention above, we plan to publicly share our denoisers pre-trained using both SN strategies with our code. Due to recent activity around training of nonexpansive neural nets, we expect further improvements in such techniques in the near future.
>
> - Similar to the AWGN denoisers, our AR operators were trained to be 1-Lipschitz using SN. The better performance of AR priors comes from their training on specific recovery tasks.  Unlike AR priors, however, the AWGN denoisers in PnP/RED are not trained for the specific measurement model, making them more general, but also limiting their ability to account for the non-AWGN artifacts across iterations [59].
>
> - All typos will be fixed in the revised manuscript.

---

### Official Review · Reviewer_rXws · 2021-07-16

**Rating:** 7
**Confidence:** 4

**Summary:**

In this work, the authors analyze Plug and Play (PnP) methods for solving inverse problems, and utilize the framework from compressive sensing with generative models to derive theoretical guarantees. In particular, the authors show that in a linear inverse problem, if 1) the residual operator of a denoiser is Lipschitz and 2) the measurement matrix satisfies a restricted eigenvalue condition over the range of the denoiser, then linear convergence of the PnP algorithm can be established for suitable parameter choices. Moreover, under a similar set of assumptions, an equivalence between PnP and Regularization by Denoising (RED) is derived, in that both algorithms have the same set of solutions. Empirical evaluations also show 1) supporting evidence for aspects of the theory holding for trained neural network-based denoisers and 2) that PnP algorithms compete with, and can outperform, state-of-the-art compressive sensing approaches using deep generative priors.

**Limitations And Societal Impact:**

The authors adequately discussed various limitations and broader impacts in Sections 5 and 6 of the manuscript.

**Main Review:**

# Originality:

The theoretical results presented in this work are, to the reviewer’s knowledge, the first to combine results related to CS with generative priors to PnP methods. The particular convergence analysis seems to be an extension of [23], by showing that the S-REC property over the range of the denoiser is sufficient for convergence, rather than the strong convexity and smoothness property of the data fidelity term as assumed in [23]. Indeed, in the appendix it is shown that S-REC is equivalent to (restricted) strong convexity of the data fidelity term. Moreover, all iterates are shown to be in the image of the denoiser (by definition of the PnP algorithm). On this note, it would be good if the authors could comment on the larger technical difference between the proof technique presented here and that of [23]. For example, does focusing on S-REC to prove convergence provide technical hurdles over [23]?

# Quality:

- Strengths: - The paper adds novel analysis to the literature on theoretical guarantees for PnP algorithms, and have the potential to spark future research into using tools from generative priors to analyze PnP algorithms. - It is well-written and easy to follow. - The theoretical guarantees provide convergence rates on the algorithm, and experimental evidence is shown for some of the theoretical assumptions to approximately hold in practice with neural network-based denoisers. - The empirical results showcase the effectiveness of PnP algorithms against state-of-the-art generative prior-based approaches.

- Weaknesses: - It is unclear when the S-REC property would hold for denoisers, or how many measurements are required for it to hold. In the case of CSGM, there is a clear connection between the “complexity” of the generative model and the sample complexity required for $A$ to satisfy S-REC (namely, $m = O(k \log L)$ random Gaussian measurements suffice for an $L$-lipschitz generator with latent dimension $k$). However, the authors did not comment on what scenarios we should expect this condition to hold for neural network-based denoisers. What sorts of notions of “complexity” could potentially hold for the range of denoisers that would provide some intuition as to when we should expect S-REC to hold? Since the image of $D$ consists of clean, natural images, we should expect it to be approximately low-dimensional. Are there aspects of their architectures that could provide some (quantitative) answers as to how low-dimensional the image of $D$ might be in practice?

# Clarity:

Overall, the paper is well-written. I found the discussion of the results and proofs easy to follow and related work seems to be cited properly. The following are a few typos found:
- pg 2 line 60: “optimizatio” -> “optimization”, pg 9 line 296: “reduced” -> “reduce”

# Significance:

I think it is interesting to draw a connection between PnP methods and generative prior-based methods, as both have some similarities in terms of their philosophy (namely, they detach learning data-driven priors from downstream inverse problems). Given the recent developments in [23] along with the combined analysis of CSGM and the present work, there’s certainly the possibility for more follow-up work from the community to provide further theoretical analyses for PnP methods using some of the tools from the field of generative priors. This could spark studies of the “complexity” of denoisers and the relationship between the architectures of denoisers and the quality of the PnP algorithm for various measurement operators. It would also be of interest to see if techniques from generative prior-based approaches for nonlinear inverse problems could then aid in deriving guarantees for PnP methods applied to nonlinear measurements.

**Time Spent Reviewing:**

4

---

> ### Author Response · Authors · 2021-08-09
> **Response to Reviewer rXws**
>
> We appreciate your thoughtful comments and positive assessment of our work. After carefully reviewing your feedback, we provide below answers to the comments you raised. In short, while our Theorem 1 builds on the analysis in [23], it additionally uses Lemma 4 in the supplementary material to link the fixed points of PnP to the solution of the inverse problem. We expect that the future work will be able to relate the number of features in the bottleneck layers of encoder-decoder-type denoisers to the number of measurements.
>
> Originality:
> - As you point out, our Theorem 1 is related to the analysis in [23]. The key points of similarity and difference between the two analysis techniques can be understood by reviewing our proof in Section A of the supplementary material. Specifically, Lemma 4 in our analysis is not present in [23]. It allows to explicitly relate the fixed points of PnP-PGM to the underlying solution to the inverse problem. Note that [23] simply establishes the fixed-point convergence of PnP without relating it to the underlying solution. On the other hand, as stated in the corresponding proofs, Lemma 3 and Lemma 6 were directly adopted from the analysis in [23]. Finally, Theorem 2 and Theorem 3 in our work do not have analogues in [23]. We will state these relationships in the corresponding sections of the revised supplementary material.
>
> Weakness:
>
> - Indeed, the next big step in the theoretical understanding of PnP/RED would be the establishment of a relationship between the number of measurements and the complexity of the denoiser. While we do not have corresponding results at the moment, we are actively working in this direction by building on the analysis in [5]. One natural notion of complexity for the range of a denoiser is the number of coefficients in the bottleneck of an encoder-decoder architecture and the number of layers. We can ensure a low-complexity range space by restricting the bottleneck to have only $k$-coefficients (with $k << n$). Fixed points of such denoisers can also be intuitively understood as images compressed to $k$-dimensional latent spaces. We will mention the discussion above in the revision as a potential avenue for future work.
>
> Clarity:
>
> - We will fix all the typos in the revised manuscript.

---

> > ### Comment · Reviewer_rXws · 2021-08-26
> > **Post-rebuttal response**
> >
> > Dear authors,
> >
> > Thank you for your response. After reading over the other reviews and rebuttals, I would like to keep my score and continue to recommend accept. I think the work is solid and will be of interest to the NeurIPS community.

---

### Official Review · Reviewer_V21i · 2021-07-16

**Rating:** 7
**Confidence:** 4

**Summary:**

This paper considers recovery guarantees for Plug-and-Play algorithms (PnP), where an image denoiser is used to solve inverse problems such as compressed sensing. Theoretical results show that under certain assumptions, the error between the estimate and ground truth decay linearly, and experiments show that the practical implementations of the proposed algorithm gives state of the art reconstructions.

**Ethical Concerns:**

No obvious ethical concerns.

**Limitations And Societal Impact:**

No obvious societal concerns.

**Main Review:**

__Strengths__:
1. This paper provides the first recovery guarantees for PnP methods in compressed sensing. While traditional results consider asymptotic convergence of the algorithms, this paper characterizes the error decay between the estimate and the ground truth.

1. The authors elaborate the results clearly. They first state a Theorem for ground truth lying in the range of the denoiser, then extend it to arbitrary signals, and finally show that the fixed points of their proposed PnP algorithm coincides with that of RED, another popular PnP algorithm.

1. The authors also propose a novel algorithm that uses PnP along with an artefact removal network. While the results have better PSNRs, I cannot see any difference in reconstruction quality.

__Weaknesses__:
1. An obvious weakness is that under assumption 1, if $ \|\| x - D(x) \|\|_2 \leq \delta $ for all $x \in \mathbb{R}^n$, then the range of $D$ will need to contain a full dimensional subset of $\mathbb{ R }^n$. This immediately implies that for S-REC to hold, the number of measurements needs to be at least $n$, and compressed sensing recovery should not be possible. This only seems to affect Theorem 2 and Theorem 3, since Theorem 1 assumes $x^*$ lies in the range of $D$ and hence does not require the previous assumption.

1. I have some concerns about the constants. If $\alpha < 1$, then assumption 1 implies that $ | x - z | - | D(x) - D(z) | \leq | x - z - D(x) + D(z) | \leq \alpha | x - z | \Rightarrow |D(x) - D(z) | \geq ( 1 - \alpha ) | x - z | $. Since $D$ needs to be both Lipschitz ( from Assumption 3) and reverse-Lipschitz (from the previous derivation), the range of $D$ must be the whole of $\mathbb{R}^n$, making this a terrible denoiser. This makes the condition $\alpha \geq 1$ necessary, which later affects the parameters $c$ in equation 9 and $\varepsilon$ in equation 12. I think there is some range of parameters that are physically meaningful, but the authors should detail this.

1. The artefact removal network seems to help the PnP method, but it is difficult to find any qualitative differences between the reconstructions with / without this extra network. Additionally, the use of this network seems somewhat auxiliary to the rest of the results.

__Novelty and significance__:
While I am not an expert on PnP methods, to the best of my knowledge this is the first result that gives a non-asymptotic recovery result wrt the ground truth $x^*$.

__Clarity__:
I found the paper easy to read, and the results were elaborated clearly.

__Score justification__:
Overall I think this is a good paper. The analysis techniques are fairly standard, but are used innovatively and make connections between existing work, for e.g., the link between S-REC and Restricted Strong Convexity. While I think assumption 1 is currently too strong, I think the rest of the analysis is novel and informative, and perhaps future work can avoid this assumption. Hence I'm in favor of acceptance.

__General comments__:
1. In line 58 and line 69, use the actual spaces $\mathbb{R}^n$ and $\mathbb{R}^n$ instead of $Im(W)$, $Im(D)$. This helps avoid confusion on the dimension of the output.

1. It is worth pointing out the scaling of the term $\lambda = \sigma_{\max} (A^T A)$. Even for sub-gaussian matrices, this can be as bad as $O(\sqrt{ \frac{ n }{ m } })$

1. Please restate the Theorem statement in the appendix / proofs.




**Time Spent Reviewing:**

3

---

> ### Author Response · Authors · 2021-08-09
> **Response to Reviewer V21i**
>
> Thank you for your feedback and positive assessment of our work. We have carefully read your comments, which will help us to improve the paper. Below, we provide individual responses to your comments. In short, in our quest for maximally compact statements, we have stated our assumptions in an unnecessarily strong fashion. Those assumptions can be significantly weakened without limiting the validity of our theoretical analysis.
>
> Weaknesses:
>
> - Indeed, as you point out, the boundedness of the residual in Assumption 1 is unnecessarily strong. The good news is that this assumption—which is only used in the proof of Theorem 2—can be significantly weakened. As can be seen in eq. (3) of the supplementary material, the constant $\delta$ is only used to bound the norm of the residual at the fixed-point of PnP-PGM (see the second inequality). Thus, for the analysis of Theorem 2 to hold, we do not need the residual to be bounded everywhere, but only at the fixed points. We will clarify this in the revised version of the paper.
>
>
> - Similarly, the nonexpansiveness of $D$—which is only used in Theorem 3—can be significantly weakened. As can be seen in line 77 of the supplementary material, we do not need $D$ to be nonexpansive everywhere, but only at the fixed points of RED-SD (see definitions of $x$ and $z$ in lines 74 and 75). Additionally, as you pointed out, we do not always need $\alpha < 1$, so long as the constant $c < 1$. We will clarify this in the revised manuscript by also providing a meaningful range for $\alpha$ to have $c < 1$, which is $\alpha < 2\mu/(\lambda-\mu)$.
>
>
> - The current view of PnP/RED methods is that they are limited to priors performing AWGN denoising. Our numerical results present an alternative view by using deep priors trained for more general artifact removal. This partially addresses a widely held misconception on PnP/RED by also highlighting that our theory is applicable to both PnP (denoising) and PnP (AR). We will include error residuals in the revised supplementary material to highlight qualitative differences between methods.
>
>
> General comments:
>
> - Following your recommendation, we will use the actual spaces.
> - We will provide $\lambda_{\max}$ values in the revised manuscript.
> - Following your recommendation, we will restate the theorems in the proofs.

---

> > ### Comment · Reviewer_V21i · 2021-08-27
> > **Follow up to author response**
> >
> > Thank you for your response.
> >
> > Based on the other reviewer concerns and the author response, I continue to remain positive about this paper. It seems that my original concerns about the strength of the assumptions can be addressed, and I strongly recommend the authors include this in the final version.
> >
> > I don't see a reason to increase or decrease my score, and it remains at 7.

---

### Decision · Program_Chairs · 2021-09-27

**Decision:**

Accept (Poster)

**Comment:**

This paper provides the first recovery guarantees for Plug-And-Play (PnP) and Regularization by Denoising (RED) methods for compressed sensing.  While it uses similar analysis as in the literature for compressed sensing from generative priors, this extension is significant because it shows those tools work for a class of methods that are computationally cheaper than generative models.  This work could be improved by a study of the conditions under which the Set-Restricted Eigenvalue Condition holds in the present context.  Despite this weakness, the work has the potential to inspire additional theoretical interest in PnP/RED for solving inverse problems.  In the camera ready, please clarify where the residual boundedness and the nonexpansiveness of D must hold, as discussed with the reviewers.